



# Spatial and temporal variability of Terminal Classic Period droughts from multiple proxy records on the Yucatan Peninsula, Mexico

Stephanie C. Hunter[1*], Diana M. Allen[1], and Karen E. Kohfeld[2]

[1]Department of Earth Sciences, Simon Fraser University, Burnaby, BC, V5A 1S6 Canada

[2]School of Resource and Environmental Management, Simon Fraser University, Burnaby, BC, V5A 1S6 Canada

*Correspondence to*: Stephanie Hunter (stephanie_hunter@sfu.ca) Department of Earth Sciences, Simon Fraser University, 8888 University Drive, Burnaby, BC, Canada, V5A 1S6

**Abstract.** The Terminal Classic Period (TCP, 800-1000 A.D.) coincides with the collapse of the Maya Civilization on the Yucatan Peninsula, a period of rapid population decline that has been attributed to extended and widespread droughts. This study uses multiple proxy records from the Yucatan Peninsula to collectively analyze drought occurrence across the region during this time. We use a changepoint analysis to identify periods of significant changes in the statistical properties (mean and variance) of 23 proxy records and classify evidence of drought based on four criteria: (1) a changepoint in mean and variance during the TCP, (2) a change towards more arid conditions during the TCP, (3) a change greater than 20% from the time-series mean, and (4) having a mean during the TCP that is significantly different from the time-series mean. Our analysis shows that five records met all inclusion criteria for showing definitive evidence of drought during the TCP, and these are located in the northwest, northeast, and north-central regions of the Yucatan Peninsula. Many of these records showed some evidence of drought (meeting some but not all criteria), but some showed evidence of drought occurring earlier than the TCP (in the northeast of the Yucatan Peninsula) and later than the TCP (in the south of the Yucatan Peninsula). We also conducted a changepoint analysis on reconstructions of three modes of climate variability known to affect the movement of the Intertropical Convergence Zone (ITCZ). Our comparison suggests that during the first half of the TCP, the Pacific Decadal Oscillation (PDO), El Niño Southern Oscillation (ENSO), and Atlantic Multidecadal Oscillation (AMO) were all in positive phase, which may have pushed the ITCZ southward during the winter months and enhanced aridity during the dry season. However, our analysis suggests that the position of the ITCZ was not the sole driver of the TCP droughts, as these conditions existed over the Yucatan Peninsula prior to the TCP as well. This study highlights the complexity of the spatial and temporal variability of these droughts, and points to the need for further study to identify the mechanisms responsible for the TCP droughts.

## 1.0 Introduction

Multiple efforts have been made to reconstruct climate of the Yucatan Peninsula during the Terminal Classic Period (TCP, 800-1000 AD), a period of reported extensive droughts which have been suggested as a significant contributing factor in the ultimate collapse of the Maya Civilization (Hodell et al., 1995; Curtis et al., 1996, 1998; Hodell et al., 2005a, 2005b; Escobar, 2010; Medina-Elizalde et al., 2010; Stahle et al., 2011, Kennett et al., 2012; Akers et al., 2016). The story of these droughts and their contribution to the demise of the Mayan people is a compelling one; the existence of droughts of this magnitude, even before the beginning of human-induced climate change, suggests that they could occur again (Cook et al., 2010). In addition, the latest report by the





Intergovernmental Panel on Climate Change (IPCC) states that under future projected climate change, precipitation
extremes (floods and droughts) will increase, and that some areas around the globe may be even more prone to
droughts (Hoegh-Guldberg et al., 2018).

The precipitation regime on the Yucatan Peninsula is associated with the migration of the Intertropical

Convergence Zone (ITCZ), a low pressure zone that circles the equator where the northeast and southwest trade
winds converge. The ITCZ is a controlling factor of rainfall distribution in the equatorial regions of the globe
(Hastenrath, 1966). The ITCZ also interacts with other climate cycles and teleconnections, such as the El Niño
Southern Oscillation (ENSO), the Pacific Decadal Oscillation (PDO), and the Atlantic Multidecadal Oscillation
(AMO) (Pavia et al., 2006).  These modes of variability can interact with each other, enhancing or muting the effects
of one another. Furthermore, as potential mechanisms for extreme drought, these climate cycles may be altered by
anthropogenic climate warming, and need to be considered in studies of the future climate (Rauscher et al., 2011;
Fasullo et al., 2018).

While numerous paleoproxies have been used to study the existence of past droughts on the Yucatan

Peninsula (Hodell et al., 1995; Curtis et al., 1996, 1998; Hodell et al., 2005a, 2005b; Escobar, 2010; Medina-
Elizalde et al., 2010; Stahle et al., 2011, Kennett et al., 2012; Akers et al., 2016), no study has analyzed the spatial
and temporal distribution of these droughts, nor  determined the driving mechanism (or combination of
mechanisms).  This study analyses 23 proxy records from the Yucatan Peninsula and surrounding region to study the
spatio-temporal patterns of drought during the TCP (Figure 1).  These records provide qualitative reconstructions of
changes in precipitation, soil moisture, or the hydrologic budget derived from records based on fossil shell $\delta^{18}O$,
lake sediment properties (magnetic susceptibility, sediment density), speleothem $\delta^{18}O$, and tree rings.  The methods
used in this study are used to help quantify the occurrence of drought, and enhance the qualitative assessments that
have been completed previously (e.g. Douglas et al., 2016).

The first quantitative estimate of precipitation reconstructed for the TCP was based on a statistical

relationship derived using an observed composite precipitation record for the Yucatan and the oxygen isotope ($\delta^{18}O$)
composition of calcite extracted from a speleothem near Merida, Mexico (Medina-Elizalde et al., 2010).  Other
proxy records available for the Yucatan Peninsula include fossil shells (Hodell et al., 1995; Curtis et al., 1996, 1998;
Hodell et al., 2005a), sediment components (Hodell et al., 2005a; Escobar, 2010), speleothem deposits (Kennet et
al., 2012; Akers et al., 2016), and tree rings (Stahle et al., 2011).  These proxies can be used to infer moisture
availability from some property of the proxy that is sensitive to evaporation and precipitation changes, such as $\delta^{18}O$,
the ratio of calcite to gypsum in sediment, or tree growth (Hodell et al., 1995; Curtis et al., 1996, 1998; Hodell et al.,
2005a, 2005b; Escobar, 2010; Medina-Elizalde et al., 2010; Stahle et al., 2011; Kennett et al., 2012; Akers et al.,
2016).  These proxies have all been used as indicators of time periods in the past which were "dry" (periods of
drought) regardless of whether the drought was caused by reduced precipitation, increased evaporation or other
climate forcings.





However, the timing, duration, and magnitude of the TCP droughts are not consistent throughout the

region. Oxygen isotope records at Punta Laguna, Yucatan (Curtis et al., 1996) show a distinct peak at $862 \pm 50$ AD
(see Figure 1), which is interpreted as the drought that led to the Mayan collapse. The peak in the Punta Laguna
record roughly coincides with the droughts suggested by the speleothem record at Tecoh Cave (Medina-Elizalde et
al., 2010), which suggests a generally arid period during the TCP with the most arid conditions occurring at 806,
829, 842, 895, 921, and 935 AD. Similarly, $\delta^{18}O$ records from Lake Chichancanab (Hodell et al., 1995) also show a
dry period during the TCP, with peak aridity at 922 AD. Sediment density and mineralogy records from this same
location suggest an increase in the frequency of droughts between the period from 750 to 800 AD (Hodell et al.,
2005a, 2005b). To the west of the Yucatan Peninsula, the reconstructed Palmer Drought Severity Index (PDSI)
using tree rings (Stahle et al., 2011) suggests below average moisture conditions at approximately 900 and 1100 AD,
in the late TCP. However, this record is located in central Mexico, and moisture conditions at this location
(Barranca de Amealco) do not seem to be correlated with that of the Yucatan Peninsula (Medina-Elizalde et al.,
2010; Stahle et al., 2011). Conversely, further south at Petén Itzá (in the Mayan Lowlands), $\delta^{18}O$ records do not
strongly support TCP droughts (Curtis et al., 1998), although there appears to be a short period of aridity around
1000 AD at the end of the TCP. A magnetic susceptibility record from Petén Itzá (Escobar, 2010) also suggests
wetter climate at the beginning of the TCP (approximately 800 AD), but the TCP is not analyzed. More recent
studies from Belize, at the south of the Yucatan Peninsula, do show some evidence of drought during the TCP,
although the strongest periods of aridity do not appear to be during the TCP (Kennett et al., 2012; Akers et al.,
2016). At Yok Balum cave, speleothem $\delta^{18}O$ records show five period of drought around 400, 900, 1100, 1580, and
1780 AD, with the strongest drought appearing to be around the year 1100 AD (Kennett et al., 2012). Another
location farther north in Belize, Macal Chasm, suggest four periods of drought since the year 400, with droughts
centered around 800, 1100, 1580, and 1900 AD. Again, the strongest drought at this location appears to be around
1100 AD (Akers et al., 2016). The differences in the timing of these droughts suggests there may be some spatial
differences in the occurrence of the TCP droughts, but these inconsistencies across studies may also be an effect of
uncertainties in the age models of these proxies. A number of the age models are constrained by only one
radiocarbon date, which increases the uncertainty in the age model (Telford et al., 2004) (discussed in Section 4.3).

Several mechanisms have been proposed to explain droughts on the Yucatan Peninsula, and there has still

been no agreement on which one (or combination) of these mechanisms was responsible for the TCP droughts.
Hodell et al. (2001) suggested that solar forcing could be responsible for the droughts, as lake sediment cores from
the northern Yucatan Peninsula showed a periodicity at approximately 200 years, matching paleoclimate records of
solar activity. Medina-Elizalde et al. (2010) also noted 200-year cycles in the Chaac speleothem record, further
corroborating the idea that solar activity may be related to moisture availability on the Yucatan Peninsula. Changes
in the strength of the North Atlantic Meridional Overturning Circulation (AMOC) have also been suggested as a
control on rainfall on the Yucatan Peninsula (Enfield and Alfaro, 1999; Giannini et al., 2000; Taylor et al., 2011).
Curtis et al. (1996) also suggested that the variability observed in the fossil shell $\delta^{18}O$ records at Punta Laguna was
caused by changes in North Atlantic atmospheric and oceanic circulation; millennial scale cycles in the proxy
records at Punta Laguna roughly matched the periodicity of North Atlantic cooling events known as Bond events



(Bond et al., 1997, 2001). Changes in the Pacific Ocean, specifically the effect of ENSO on tropical precipitation,
also could have caused the TCP droughts (Lachniet et al., 2004; Lachniet et al., 2012). Wahl et al. (2014) also
suggested that increased ENSO variability in the late Holocene could be related decreased precipitation in Central
America. However, there have been no records of past ENSO that indicate ENSO was particularly frequent during
the TCP, suggesting it was not the sole cause of droughts on the Yucatan Peninsula (Douglas et al., 2016).

A couple of hypotheses put forward as potential drivers of Yucatan droughts have since been rejected.

Medina-Elizalde and Rohling (2012) suggested that reduced tropical cyclone activity contributed to drought
conditions during the TCP, bringing less rainfall in the form of storms. Frappier et al. (2014) tested this hypothesis
by analyzing mud layers in cave deposits on the Yucatan Peninsula, which were interpreted as cave flooding events
brought about by storms. However, the evidence did not support the theory of reduced tropical storm activity during
the TCP. Deforestation has also been suggested as playing a role in the TCP droughts (Oglesby et al., 2010; Cook et
al., 2012), but pollen studies suggest that deforestation occurred around 800 years before the TCP (Leyden 2002).
Therefore, deforestation alone could not have caused the TCP droughts, although it may have played a role in
enhancing the droughts (Cook et al., 2012).

The final proposed mechanism for these droughts, and the one that is further explored in this study, is the

southward migration of the Intertropical Convergence Zone (ITCZ), a convective belt of low pressure which roughly
lies over the equator (Hastenrath, 2002). Historically, precipitation on the Yucatan Peninsula has been strongly
controlled by the ITCZ on seasonal and annual time scales. Total annual precipitation on the Yucatan Peninsula
ranges from 840 to 1500 mm/year, with 75 % of that rainfall occurring during the wet season (May to October). At
this time of year, the Yucatan Peninsula lies close to the northernmost extent of the ITCZ and therefore receives
greater amounts of rainfall. In the winter (November to April), the ITCZ migrates southward and the Yucatan
Peninsula experiences its dry season (Haug et al., 2001, 2003; Hodell et al., 2005b; Medina-Elizalde et al., 2010).
These shifts in the ITCZ also control the annual variability in rainfall amounts, and there is evidence that the ITCZ
shifts on a longer time scale as well, bringing centuries-long wet and dry regimes to the Yucatan Peninsula (Haug et
al., 2003). These shifts in the ITCZ could therefore be linked to the existence of the TCP droughts on the Yucatan
Peninsula, and the location of the ITCZ relative to the peninsula could explain differences in the timing and
magnitude of these droughts among the various proxy records from this region.

One method of identifying differences in behavior within a time series is via a changepoint analysis

(Killick et al., 2010), which can be used, for example, to identify points of abrupt change associated with shifts in
precipitation regimes. We posit that shifts in the position of the ITCZ will result in a corresponding shift in the
moisture availability on the Yucatan Peninsula, which will be recorded as a "discontinuity" or abrupt change in the
proxy records that are sensitive to changes in climate. A discontinuity occurs where there is a significant and rapid
change in the statistical properties of a time series, such as the mean, variance, or trend (Killick and Eckley, 2014;
Beaulieu et al., 2012). There has been growing interest in identifying regime shifts in time series from around the
world, including those of climate parameters (Tomé and Miranda, 2004; Reeves et al., 2007; Beaulieu et al., 2012),
extreme events (Zhao and Chu, 2009; Sarr et al., 2013), hydrometeorologic parameters (Seidou et al., 2007),

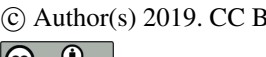



oceanographic parameters (Killick et al., 2010), atmospheric temperature (Jandhyla et al., 2013), and paleoclimate
time series (Trauth et al., 2018).

The goal of this study is to statistically analyze multiple paleo-records, including tree rings, speleothems, as

well as gastropods, ostrocods and mineralogical changes found in lake sediments, to identify periods of drought
during the Terminal Classic Period (TCP, 800-1000 AD) on the Yucatan Peninsula (Figure 2). This work is novel in
that we combine multiple types of proxy records (as well as numerous records) and analyse them collectively to (a)
minimize uncertainties inherent to individual proxy records, (b) identify any regional patterns, and (c) explore
temporal shifts in drought occurrence during the TCP. The results of our analysis are then used to discuss why these
differences might exist. This study also examines the limitations and uncertainties involved with reconstructing past
drought using different types of proxy records.
**2.0 Data and Methods**
**2.1 Types of proxy records and their limitations**

Multiple proxy types (23 in total) from eight sites on the Yucatan Peninsula were analyzed (Supplementary

Information S1 and Table S1).
There are numerous sources of uncertainty in proxy records, including the constructed age model (Ohno et al., 1993;
Kohfeld and Harrison, 2000; Telford et al., 2004), which is affected by radiocarbon dating (Stuiver et al., 1980;
Breitenbach et al., 2012); anomalous ages due to the "freshwater reservoir effect" (Broecker and Walton, 1959;
Philippsen, 2013); and changes in the rate of sedimentation/growth of the proxy, which are smoothed out by linear
interpolation methods necessary for age modeling (Boers et al., 2017). The various age models for these proxies
were assessed based on the number of radiocarbon dates used to create the age model and whether other high
resolution dating techniques were used (for example, ring counting for the tree ring record). While assessment of
the age models was not used to exclude any records from this analysis, it did help to determine which records may
be subject to more uncertainty in the age model. Uncertainties related to the age model are discussed in Section 4.3
(Limitations and Uncertainty).

For $\delta^{18}O$ proxies specifically (ostracods/gastropods in lake sediments and speleothems), the relationship

between $\delta^{18}O$ and climate is based on the assumption that carbonate forms in equilibrium with ambient water (lake
water for shells or cave drip water for speleothems). The composition of the water is controlled by multiple climatic
factors, such as temperature, evaporation, relative humidity, and precipitation (Lachniet, 2009). Precipitation into the
lake can change the lake composition, as precipitation over the Yucatan Peninsula (storms, in particular) is depleted
in $\delta^{18}O$ compared to lake water (Hodell et al., 1995; Lawrence and Gedzleman, 1996; Perry et al., 2003); the size of
the lake acts to control the extent to which the $\delta^{18}O$ composition of precipitation affects the composition of the lake,
while temperature, humidity, and evaporation act to control the amount of water evaporating off of the lake.
However, for an idealized closed system, the composition of the water is assumed to only reflect the inputs and



outputs of the systems, which would be precipitation (P) and evaporation (E), respectively, and the composition of
the lake and temperature (as well as humidity) are assumed to be constant (Hodell et al., 2005a). Therefore,
recognizing that different species of $\delta^{18}$O proxies respond differently to temperature, evaporation, and humidity,
(Holmes and Chivas, 2002), we interpret increases in $\delta^{18}$O as increases in the amount of evaporation relative to the
amount of precipitation (dry periods), while decreases in $\delta^{18}$O are interpreted as an enhancement in precipitation
input into the lake (wet periods).

The speleothem record used in this study was considered to reflect mean annual precipitation due to the

high correlation of $\delta^{18}$O with an observed composite precipitation record from the Yucatan Peninsula (Medina-
Elizalde et al., 2010). Because the average annual temperatures in the tropics have been relatively stable over the
Holocene (temperature reconstructions by Marcott et. al., (2013) estimate tropical temperature changes to be ~ 0.4°C
over the early Holocene, followed by little to no temperature changes in the late Holocene), the variability of $\delta^{18}$O
was assumed to be entirely attributed to changes in the ratio between evaporation and precipitation, or E/P (and not
to changes in temperature alone). However, the assumption of stable temperatures during proxy formation is a large
uncertainty which carries through to the other $\delta^{18}$O proxies as well. The relationship between the fractionation
factor (for calcite and water) and temperature suggests that for every degree decrease in temperature, there is a
~0.2‰ increase in $\delta^{18}$O (Kim and O'Neil, 1997). Because dry periods are recorded in these proxy records as
enrichment in $\delta^{18}$O, the apparent periods of drying in the proxy record could be partly attributed to cooler periods
that occurred throughout the proxy record. For example, a 1°C cooling would result in an apparent decrease in
precipitation of 48 mm/year in the speleothem record used in this study (Van Pelt, 2016). This relationship would
be different for all $\delta^{18}$O proxy records, highlighting their uncertainty, especially those with a low range in $\delta^{18}$O (1-
2‰). Previous speleothem studies have suggested that during the past 2000 years, temperatures may have been up
to 0.4°C cooler than today (Marcott et al., 2013), which would result in an apparent decrease in precipitation of 19
mm/year. With annual precipitation on the Yucatan Peninsula ranging from 840-1500 mm/year on average
(Gondwe et al., 2010), this apparent decrease in precipitation could account for up to 2.2% of any changes in
precipitation observed in the proxy records.

Additional uncertainty is introduced into the proxy records due to time lags that may occur duration proxy

formation. Time lags exist due to the physical processes that take place as the proxy forms; for example, a
speleothem is formed from cave drip water, and its oxygen isotope composition may be partially related to the
amount of precipitation outside the cave. However, it may take a long time for this precipitation to reach the cave -
it has to infiltrate through soil and rock in order to reach the cave, causing a time lag in the relationship between
speleothem oxygen isotope composition and precipitation. This time lag may be on the order of multiple years in
cave systems; $^3$H-$^3$He dating of cave drop waters by Kluge et al. (2010) suggested that the lower limit of cave transit
times was between 2-4 years, with this time only accounting for the transit time of water through the saturated part
of the subsurface. Medina-Elizalde et al. (2010) found that the Tecoh Cave speleothem $\delta^{18}$O was lagged behind
precipitation by 6 years using a cross correlation analysis. Time lags further complicate interpretation of the
proxies, especially when comparing the timing of drought in different records. For these reasons, a multi-proxy





approach may strengthen confidence in interpreting the changes in multiple proxy types, because identifying a
common climate signal in multiple proxy types would suggest that the signal is not a result of the assumptions or
uncertainties related to the proxy itself.

Mineralogical type proxies relate the proportions of different minerals in lake sediments to changes in

climate parameters.  Both the relative abundance of calcite (%$CaCO_3$) records (Hodell et al., 1995) and the sediment
density record (Hodell et al., 2005b) are related to changes in E/P, with lower %$CaCO_3$ and greater sediment density
signaling a higher E/P ratio.  Gypsum ($CaSO_4$) and calcite ($CaCO_3$) are the dominant minerals in lakes, with lesser
amounts of celestite ($SrSO_4$), aragonite ($CaCO_3$) and dolomite ($CaMg(CO_3)_2$).  In a closed lake system saturated
with these minerals (as is the case for Lake Chichancanab), the inputs and outputs to the system are only E and P.
Changes in the E/P ratio cause the lake volume to increase or decrease, controlling the amount that each mineral
precipitates in the lake sediments.  A high E/P ratio causes gypsum to precipitate, causing a decrease in %$CaCO_3$
(Hodell et al., 1995).  Also, because gypsum is denser than calcite, gypsum-rich layers formed during periods of
high evaporation relative to precipitation, also have higher sediment density (Hodell et al., 2005b).  Magnetic
susceptibility of lake sediments can also be used to infer changes in E/P in a closed system.  Escobar (2010) used
magnetic susceptibility to show alternating clay and gypsum units in the sediment, which corresponded to both wet
and dry periods, respectively.  Clay has a high magnetic susceptibility, so an increase in the E/P ratio was inferred in
this record by a decrease in magnetic susceptibility (Escobar, 2010).

Finally, tree ring widths (Stahle et al., 2011) are thought to record changes in drought. Tree ring widths

have been related to changes in the June Palmer Drought Severity Index (PDSI), a measure of aridity that is used to
quantify periods of drought (Palmer, 1965). The PDSI uses the difference between moisture supply (precipitation)
and water demand (in the form of evapotranspiration) to calculate a standardized PDSI value. A drier climate in the
PDSI record is indicated by decreasing, negative values of PDSI, and wetter climates are indicated by higher,
positive values for PDSI. Tree growth is sensitive to changes in both temperature and precipitation, and therefore to
evapotranspiration, which allows for a strong correlation between tree growth and PDSI.  However, the PDSI
record used in this study is not located directly on the Yucatan Peninsula, but instead in central Mexico at Barranca
de Amealco (see Figure 1), and so it may not record the same climatic conditions as on the Yucatan Peninsula.
Previous studies have shown that the PDSI record from Barranca de Amealco is not correlated to the climate on the
Yucatan Peninsula (Medina-Elizalde et al., 2010; Stahle et al., 2011).  Nevertheless, this record is included in this
analysis for comparison purposes to aid in the spatio-temporal investigation of drought in this region.
**2.2 Changepoint analysis**

The 23 records (Table S1) were analyzed to detect significant changepoints in each time series using the

changepoint package in R.  A changepoint analysis detects points in a time series at which the statistical properties
change; in this case, the statistical properties that change are the mean and variance (Killick and Eckley, 2013).  A
change in both the mean and variance is assumed to indicate a change in the climate, as each proxy record may be
responding to different (or multiple) climate variables.



A number of methods can be used to calculate multiple changepoints (described in Killick and Eckley,
2014, and references therein).  The general equation for calculating multiple changepoints is:
$$\sum_{i=1}^{m+1}\left[C\left(y_{(\tau_i+1):\tau_i}\right)\right] + \beta f(m)$$                (Equation 1)
where *C* is a cost function for a segment of the time series (with each segment separated by a changepoint, *m*, at a
certain position in the time series, $\tau$), and *ßf(m)* is a penalty value which is added to avoid over fitting of the data
(Killick et al., 2012).  Numerous functions are available for defining the penalty values and minimizing Equation 1.
For this analysis, the Pruned Exact Linear Time (PELT) algorithm (Killick et al., 2012) was chosen due to its ability
to accurately calculate changepoints while being computationally more efficient than other methods (Killick et al.,

2012).

The penalty value in Equation 1 acts to adjust the sensitivity of the changepoint analysis (decreasing the
penalty increases the sensitivity of the analysis).  While multiple options exist for the penalty functions, the choice
of penalty is somewhat arbitrary and often chosen by trial and error (Killick and Eckley, 2014).  For this analysis,
the default penalty (log(n), where n is the number of observation points) was used initially; if this resulted in too
many or too few changepoints, the penalty was adjusted manually to find a "reasonable" number of changepoints for
each time series. What is considered a "reasonable" penalty value is still undetermined, as penalty values can vary
depending on the number of data points and the magnitude of the changes; Killick and Eckley (2014) state that
currently the best practice is to plot the data with a chosen penalty and visually analyze the results to see if they
seem "reasonable". We consider a "reasonable" number of changepoints to be at least one change point, while
trying to avoid very short segments between each changepoint- this is to avoid a common problem in changepoint
analysis, which is artificially creating false changepoints by using a penalty value that is too sensitive (Killick and
Eckley, 2014).  Manual penalty values used in this analysis ranged from 0.5 to 1000, and varied based on the
number of available data points available for the period of interest and the variance of the time series.  It should be
noted that this R package in unable to incorporate missing values.  Many of the proxy records have years with no
values, so these years were deleted prior to analysis (see Supplementary Information S2).
The changepoint analysis was run twice for each proxy record: 1) to identify changes in the mean; and 2) to
identify changes in variance.  Each proxy record was then examined to determine if changes in the mean and
variance occurred during the TCP, and whether these changes were negative, which could be related to a
"meteorological drought". Meteorological drought is defined as an extended period of time without significant
precipitation (Wilhite and Glantz, 1985). This definition of drought has two components: moisture availability (in
the form of precipitation), and the length of time this moisture deficit is observed.  The changes in the proxy record
all record moisture availability in some way, either through a decrease in precipitation, an increase in evaporation, or
both (an increase in the E/P ratio).  Therefore, when notable "negative" (in the direction of less moisture
availability) changes are observed in the proxy record, they are assumed in this study to be correlated with
meteorological drought.





A 20% change in the mean value of the time series was chosen to identify notable changepoints, based on
the uncertainty inherent in $\delta^{18}O$ proxy records. As discussed above, for $\delta^{18}O$ proxy records, cooler periods could
cause an apparent drier period in the record. For proxy records with a low range in variability (e.g. 1‰), this
corresponds to an uncertainty of up to 20%, based on a 1°C decrease in temperature causing an apparent increase in
$\delta^{18}O$ of 0.20‰. Therefore, changes greater than 20% are greater than this uncertainty. This margin of error was
applied to other proxy records as well to identify the most notable changes in the proxy records. Changepoints
related to variance were also considered in this analysis as additional evidence for the existence of drought
conditions during the TCP, as the variance of the time series describes the "spread" of values from mean conditions,
and therefore is related to the existence of extreme values.
In addition to the changepoint analysis, a student's t-test was used to identify if the mean of the proxy
record during the TCP was significantly ($p < 0.05$) different from the mean of the entire time series. This additional
step was carried out to further show whether the TCP was indeed significantly different from the rest of the time
series. A two-tailed t-test assuming unequal variances was used, as the changepoint analysis showed that most of
the proxy records (16 out of 21) had a change in variance during the TCP.
The second component of meteorological drought is the length of time without moisture; the "extended
period of time" aspect of drought is incorporated in this study through the choice of penalty in the changepoint
analysis. Making the changepoint analysis less sensitive allows for the maximum segment length between
changepoints and ensures that the changepoints represent changes that existed for multiple years at a time.
In summary, the following four criteria were used to identify drought conditions during the TCP:
1.  At least one changepoint (for both mean and variance) must be present at the beginning of, during, or

immediately following the TCP to indicate that a change in the proxy record's statistical properties

occurred at that time.

2.  The change in the mean must be greater than a 20% to account for uncertainty in the proxy record.
3.  The mean of the proxy record during the TCP must be significantly different ($p < 0.05$) from the mean of

the entire time series, as calculated by the two-tailed t-test.

4.  The direction of the change in the mean must be in the direction of drying (i.e. an increased in the E/P ratio

is inferred). For some records, a decrease in the proxy value indicates drying, while for others, a drier

climate is recorded as an increase. Hereafter "positive" changes indicate an increase in moisture

availability during the TCP, while "negative" changes indicate decreases in moisture availability.

Each of these criteria is considered to provide evidence of drought during the TCP; however, the records which meet
all four of these conditions show the most definitive evidence of drought.
To provide some context for the pattern of drought variability observed in this study, we analyzed three
reconstructions of modes of climate variability (ENSO, PDO, and AMO) from Mann et al. (2009) using the same
changepoint analysis technique used to analyze the proxy records to look for patterns which may have contributed to



TCP aridity. The reconstruction of Northern Hemisphere ITCZ displacement from Lechleitner et al. (2017) is also
compared to these records. The reconstructions of the ENSO, PDO and AMO records were chosen for comparison
to the Yucatan Proxy records because all three modes of variability have the potential to affect Yucatan
precipitation, and because these records all cover the time period of the TCP.
**3.0 Results**
**3.1 Changepoint analysis**

Figure 3 shows examples of the changepoint analysis results; the remaining graphs can be found in the

Supplementary Information, Figure S1 and S2. As mentioned above, the changepoint function in R is unable to
accommodate missing years of data, and as all of the proxy records have different resolution (ranging from yearly to
decadal in scale), each data point was assigned an index number for the analysis, with 0 being the most recent data
point, and increasing index numbers going back into the past. Thus, the TCP results appear to be of different
lengths. In Figure 3, the width of the orange shaded area simply reflects the number of data points that fall within the
TCP (see Figure 3 and Supplementary Information S2).

Many records have changepoints during (or slightly before/after) the TCP; eight records show a greater

than 20% change in the mean (with four records showing a change less than 20% from the mean) and 18 records
show changes in variance during the TCP (Figure 4). Of the records that show changes in the mean during the TCP,
seven are negative (indicating drought conditions) and five are positive (indicating increased moisture availability).
One of the records with a positive change is from the south of the Yucatan Peninsula at Macal Chasm, two are from
the south at Lake Petén Itzá, one is from the northwest at Aguada X'caamal, and one is from the northeast at Lake
Chichancanab. The two-tailed t-test shows that 10 of the records have a TCP mean that is significantly different
from the mean of the entire time series. Only five records meet all four of the inclusion criteria (indicated with a star
in Figure 4).

The five proxy records that met all of the four inclusion criteria for evidence of drought were found at Lake

Chichancanab (*Cyprinotus sp.* $\delta^{18}$O, Hodell et al., 1995; and sediment density, Hodell et al., 2005b) in the north-
central Yucatan Peninsula, Punta Laguna in the northeast (*Pyrgophorus coronatus*, Curtis and Hodell, 1996;
*Cyhteridella ilosvayi*, Curtis and Hodell, 1996), and Tecoh Cave (Chaac speleothem $\delta^{18}$O, Medina-Elizalde et al.,
2010) in the northwest. These three sites show the most evidence for drought, although the proxy records are not
unanimous at one site (Lake Chichancanab).

At Tecoh Cave there was only one record (the Chaac speleothem record), but it shows that during the TCP

the mean precipitation was quite variable; all of the mean changes during this time were negative, and there are short
periods where the mean changes are greater than 20%. This record has a higher resolution during the TCP than the
other proxy records (i.e. it has annual resolution during the TCP), which likely explains why it shows greater



variability at this time. This record also shows a change in variance during and following the TCP, and offers strong
evidence of TCP droughts in the northern areas of the Yucatan Peninsula.
At Lake Chichancanab, even though four of the five proxy records show large (>20%) negative changes in
the mean around the TCP (all except for *Physocypria sp.* $\delta^{18}$O, Hodell et al., 1995), only two of its records met all
four of the inclusion criteria. Two of the Lake Chichancanab records (*Pyrgophorus sp.* $\delta^{18}$O, Hodell et al., 1995;
and %CaCO$_3$, Hodell et al., 1995) did not meet all inclusion criteria because no changes in variance near the TCP
were identified and they did not pass the t-test for significant change in mean during the TCP. This suggests these
two records may be recording drought at Lake Chichancanab, but the signal is not as strong as in the *Cyprinotus sp.*
$\delta^{18}$O (Hodell et al., 1995) and sediment density (Hodell et al., 2005b) records. The *Physocypria sp.* $\delta^{18}$O (Hodell et
al., 1995) record does show strong evidence of change during the TCP, but rather a positive change in the mean
slightly after the TCP. As this is the only record that shows a trend towards more moisture availability during the
TCP, it is possible that there is some interference with this proxy and that it is not recording changes in E/P at Lake
Chichancanab like the other proxies, or that these differences are due to uncertainty in the age model. As a whole,
the proxies at Lake Chichancanab do suggest evidence of drought, although they are a bit less certain than at Tecoh
Cave due to the lower certainty in the age model (see Section 4.3).
At Lake Punta Laguna, both of the records (*Pyrgophorus coronatus* $\delta^{18}$O and *Cytheridella ilosvayi* $\delta^{18}$O,
Curtis and Hodell, 1996) technically did not have a changepoint during the TCP, but they both have a changepoint in
the mean directly before the TCP. Both records have a changepoint in the variance at the beginning of the TCP, and
the t-test showed that they had a significantly different mean during the TCP than during the rest of the time series.
The changepoint analysis graphs (see Supplementary Information, Figure S1) show that the mean during the TCP
was significantly lower than the time-series mean leading up to the TCP and during the TCP. Therefore, these
records were included in the final count as it is possible that the placement of the changepoint prior to the TCP could
be due to uncertainty in the age model. However, if correct, this timing could suggest onset of drought in the
northeast of the Yucatan Peninsula prior to the TCP.
None of the records from Aguada X'Caamal (in the northwest region of the Yucatan Peninsula) met the
four inclusion criteria for drought in this analysis. The t-test analysis identified four out of nine proxy records at
Aguada X'Caamal that demonstrated significant differences in the mean during the TCP relative to the rest of the
record. These include the three Chara $\delta^{18}$O (algae) records and *Pyrgophorus coronatus* 2 $\delta^{18}$O (Hodell et al. 2005a).
Of these records, only the Chara 4 $\delta^{18}$O record (algae) showed a changepoint in the mean during the TCP, but this
record shows a positive change slightly before the TCP. In addition, the changes in variance for the Aguada
X'Caamal records differ among all of the records. These records do show changepoints in variance around the TCP,
although there does not appear to be any consistency in when these changepoints occur in these records
(Supplementary Figure S2). These differences between records suggest no consistent changes in variance among
the proxy records in this location. This suggests the TCP droughts may not have extended to the west side of the
Yucatan Peninsula. This corroborates the findings of Hodell et al. (2005a), who found that Aguada X'Caamal $\delta^{18}$O



values varied much more than those at Lake Chichancanab in the 15th century, and that Punta Laguna showed
opposite trends to Aguada X'Caamal.  This inverse relationship could very well be true during the TCP as well.
However, it is possible this inconclusive result is also be due to uncertainties in the age models for these proxy
records.  All proxy records at Aguada X'Caamal were oxygen isotope type proxies, and there are no other proxy
types available that can be used to verify the results.  The only other sites that have only one type of proxy are the
ones with speleothem records (Tecoh Cave, Macal Chasm, and Yok Balum Cave). These have age models with
more age ties, and therefore are more certain than the ones at Aguada X'Caamal (see Discussion Section 4.3).

The locations in the south of the Yucatan Peninsula, Lake Petén Itzá and Yok Balum Cave, as well as the

location to the west of the Peninsula, Barranca de Amealco, also show no conclusive evidence of drought during the
TCP.  Of the proxy records from Lake Petén Itzá (*Cytheridella ilosvayi* $\delta^{18}$O, Curtis et al., 1998; *Pyrgophorus*
*coronatus* $\delta^{18}$O, Curtis et al., 1998; and magnetic susceptibility, Escobar, 2010), the two $\delta^{18}$O proxies show
changepoints in the mean and variance during the TCP; however, the changepoints are from nearly average moisture
conditions to above average moisture conditions during the TCP.  This is supported by the t-test, which did not
detect significant differences in the mean in either record.  The magnetic susceptibility record, also at Lake Petén
Itzá, shows no significant change in mean or variance during the TCP, suggesting that this southern location was not
subject to the same climatic conditions as those experienced in the northern part of the Yucatan Peninsula.  Today,
the seasonality of precipitation at Lake Petén Itzá is known to be different from the northern Yucatan Peninsula,
with the dry season occurring from January-May instead of from November-April as it does farther north.

Also in the south, the speleothem record at Yok Balum Cave only met one of the inclusion criteria (a single

changepoint in variance around the TCP), suggesting it did not experience drought at this time.  The third site in the
south, Macal Chasm, appears to show some evidence for drought during the TCP, as it has a mean changepoint with
a 20% change in mean during the TCP and a change in variance; however, the changepoint is actually a positive
one, again suggesting that the proxy records in the south did not experience droughts during the TCP.  To the west
of the Yucatan Peninsula, the PDSI reconstruction from Barranca de Amealco tree rings (Stahle et al., 2011) also did
not show any significant changes in mean or variance during the TCP.  This changepoint analysis corroborates the
findings in both Stahle et al. (2011) and Medina-Elizalde et al. (2010), which showed that there is no correlation
between precipitation on the Yucatan Peninsula and at Barranca de Amealco.

Two records in the south (the Macal speleothem record, Akers et al., 2016; and the Yok Balum speleothem

record, Kennett et al., 2012) show some evidence that droughts in the south of the Peninsula, but after the TCP.
Both records have a changepoint in the mean at about 1100 A.D., indicating a change from above average moisture
conditions to below average moisture conditions.

The results of the mean changepoint analysis for the three climate variability records (AMO, ENSO and

PDO) are shown in Figure 6.  These graphs show that changepoints occur at approximately 1400 A.D. in the AMO
record, approximately 900 A.D., 1300 A.D., and 1900 A.D. in the ENSO record, and approximately 1100 A.D. in
the PDO record.  Both AMO and PDO do not have changepoints that occur near the TCP.






## 4.0 Discussion

### 4.1 Was migration of the ITCZ the cause of TCP droughts?

Previous studies have attempted to explain or allude to the possible causes of the collapse of the Mayan

Civilization; even knowing that the TCP was a period of significantly different climate, the potential mechanisms for
causing these droughts is still not certain. Possible mechanisms have included reduced tropical cyclone frequency,
increased solar activity, deforestation, increased ENSO variability, and migration of the ITCZ. The changepoint
analysis in this study explores the possible explanations that the droughts occurred in response to movement of the
ITCZ by analyzing the teleconnections that are related to its movement (Pavia et al., 2006).

On the Yucatan Peninsula, a warm ENSO phase (El Niño) is typically expressed as higher amounts of

annual precipitation, with slightly cooler temperatures during the Northern Hemisphere winter, which is typically
the dry season on the Yucatan Peninsula (November to April) (Pavia et al., 2006). However, El Niño events also
cause shifts in the timing of precipitation due to a southward shift of the winter ITCZ position, so that more rain falls
during the wet season (May to October) and less falls during the dry season. This change in timing of precipitation
can cause mid-winter droughts, despite an overall annual increase in precipitation during El Niño years (Bravo
Cabrera et al., 2010). The PDO has a similar effect on Yucatan climate, but on a longer time scale (the PDO is
described as a regime change lasting 20-30 years, compared to ENSO which has a period of 2-7 years). The PDO
and ENSO can interact by enhancing or cancelling out the effect of the other, so that the Yucatan Peninsula would
experience enhanced dry season aridity if both the PDO and ENSO are in a positive (warm) phase (Pavia et al.,
2006). Finally, the AMO acts on a longer timescale than the PDO (60-90 years), and is characterized by changes in
North Atlantic sea surface temperatures, but has widespread climate effects around the Atlantic Ocean (Schlesinger
and Ramankutty, 1994; Kerr, 2000). Knudsen et al. (2011) noted that precipitation on the Yucatan Peninsula
appears to be inversely related to variations in AMO, so that positive AMO indices (warm conditions in the North
Atlantic) correspond to less precipitation on the Yucatan Peninsula.

The AMO record shows that up until the changepoint at around 1400 A.D., the mean AMO index was in

positive phase. The corresponding effect on the Yucatan Peninsula would be a drier climate overall. In addition, the
mean PDO index remained in a positive phase until 1100 A.D., when it switched to a mean negative index. This
would also indicate drying over the Yucatan Peninsula during the TCP. Conversely, analysis of the ENSO record
shows a change in the mean ENSO index about midway through the TCP. At this changepoint, the mean ENSO
index transitioned from slightly positive to slightly negative for the remainder of the TCP. This indicates that at
least at the beginning of the TCP, the Yucatan Peninsula could have experienced enhanced winter aridity due to the
effects of ENSO. However, all of these modes of variability were also (on average) in a positive phase for at least
500 years prior to the TCP, with only ENSO having a changepoint during the TCP. This suggests that these modes
of variability were not a direct trigger for the TCP droughts, and therefore that the migration of the ITCZ due to
these modes of variability was not the sole driver of these droughts. This result is corroborated by a recent





reconstruction of Northern Hemisphere ITCZ position (Lechleitner et al., 2017), which suggests that the position of
the ITCZ was relatively stable from 0 to 1320 A.D.  Therefore, the theory of ITCZ movement alone causing the TCP
droughts is not supported by this analysis.  It is likely that ITCZ position played a role, as the three modes of
variability all suggest enhanced aridity during the TCP, but that other factors contributed to the extreme drying that
is thought to have occurred during the TCP.

**461  4.2 Were the droughts spatially and temporally variable?**

The results of this study suggest that there was spatial variability in the occurrence of the TCP droughts;
three sites in the north region of the Yucatan Peninsula show evidence of TCP drought, while the records from the
south and the northwest have inconclusive evidence for droughts during the TCP.  However, two locations showed
evidence that these droughts may have occurred at different times in different areas.  Evidence in the south of the
Yucatan Peninsula suggests that the droughts may have occurred after the TCP, and in the northwest of the Yucatan
Peninsula the proxy records suggest that the droughts may have begun before the TCP and continued into the TCP.
Lechleitner et al. (2017) notes an intense dry period around 1100 A.D. in the speleothem record at Yok
Balum Cave (Kennett et al., 2012).  This dry period was not identified by the changepoint analysis in this study,
likely because it is brief and closely followed by a period of increased moisture.  The nearby Macal Chasm
speleothem record (Akers et al., 2016) does show a changepoint at around 1100 A.D., although the mean $\delta^{18}O$ value
at 1100 A.D. is just slightly below average.  Both of these southern records, on average, do not vary too far from the
mean $\delta^{18}O$ values of these time series.  However, short and seemingly intense dry periods exist in both records.  This
evidence, combined with the possibility of earlier droughts suggested at Lake Punta Laguna, supports the idea that
droughts may have occurred at different times on the Yucatan Peninsula.  It is possible that these short and intense
dry periods at around 1100 A.D. were not as apparent at Lake Petén-Itzá (the other southern Yucatan Location)
because the proxies at this location are lower resolution than the speleothem records found at Yok Balum Cave and
Macal Chasm (see Supplementary Information, Table S1).
The other location that showed evidence of spatial variability was Punta Laguna in the northwest Yucatan
Peninsula.  Both records show a changepoint in the fossil $\delta^{18}O$ records slightly before the TCP, with below-average
moisture conditions continuing into the TCP and even after the TCP.  While it is possible that this difference in the
timing of drought onset could be a result of the age model, these records were determined to have an adequate
number of age tie point within the period of record.  Therefore, it appears that this evidence supports the theory that
droughts on the Yucatan Peninsula were varied, both spatially and temporally.  This highlights how complex these
TCP droughts were, and further emphasizes the need to further study these droughts to help better understand the
mechanisms that caused them.

**487  4.3 Limitations and uncertainty**

Our results suggest spatial and temporal variability in the timing of the TCP droughts throughout the
Yucatan Peninsula, but to a certain extent, the assessment of this timing depends on uncertainties associated with
age models generated for each record.  Therefore, the age models of this study were assessed to determine the





records that have the least and most uncertainty.  More confidence should be placed in the reconstructions with more
age ties to constrain the age model, as well as higher resolution during the period of interest; this logic is already
used by paleo-databases to rank the reliability of age models (Street-Perrott, et al., 1989; Farrera et al., 1999; Pickett
et al., 2004; Kohfeld et al., 2013).  In Figure 1, age dates are marked with stars on the time-series plots of each proxy
record, and in Figure 4 the number of age ties for the period of interest (400 A.D. to the present) and the relative
confidence in the age model are listed for each proxy.  Five records are constrained by only one age date: the
*Cyrpinotus sp.* $\delta^{18}$O, *Physocypria sp.* $\delta^{18}$O, *Pyrgophorus sp.* $\delta^{18}$O, and %CaCO$_3$ records (Hodell et al., 1995).  These
records would have the highest uncertainty, and while they do indeed collectively suggest evidence of drought at
Lake Chichancanab, there is one record (*Physocypria sp.*) that suggests increased moisture availability.  Fortunately,
the sediment density record at Lake Chichancanab (Hodell et al., 2005b) provides a separate account of moisture
availability at that location, and helps to provide more confidence in the drought observed there as it is constrained
by nine age dates.  The magnetic susceptibility record (Escobar, 2010) are also not constrained by any radiocarbon
dates; this record also did not show any conclusive evidence of drought, which could be a result of the age model
but also of its location on the south of the Yucatan Peninsula.  The four records with the most confidence are the
speleothem records (Medina-Elizalde et al., 2010, Kennett et al., 2012; Akers et al., 2016), and the June PDSI record
(Stahle et al., 2011), which was based on a tree ring chronology and annual dating using crossdating techniques.  All
other records were considered to have adequate age models, with between four and nine age dates constraining
them.  Of the records with the highest age model confidence, only one met all of the inclusion criteria for drought in
this study, and three showed inconclusive evidence of TCP drought (although the speleothem records at Yok Balum
Cave and Macal Chasm suggest a different time of drought than the TCP).

This study used a multiproxy approach to account for uncertainties inherent in different types of proxy

records.  In addition, one of the inclusion criteria for this study is that any changes in the mean must be greater than
20% deviation from the mean of the entire time series; this criterion is based on the temperature effect on oxygen
isotope proxies, but is applied to all proxy types to provide a consistent range of uncertainty. Performing a t-test
(assuming unequal variances) also serves the purpose of helping to identify changes with the highest confidence.
However, the t-test identified significant changes in the TCP mean (relative to the entire time-series mean) in proxy
records that had no changepoints during the TCP; therefore it seems useful to use both methods together.  In
addition to the inclusion criteria, the sensitivity of the changepoint identification was varied to identify the smallest
number of changepoints possible.  Choosing the smallest number of changepoints (while avoiding no changepoints
or very small segments between changepoints) allows the highest certainty possible in these changepoints; in other
words, we can be certain that these changepoints were not identified only due to the choice in penalty (by making
the analysis too sensitive).  The manually adjusted penalty values for each record are given in the Supplementary
Information (Figures S1 and S2).  Where no penalty is recorded, the default penalty value was used (log(n)).



## 5.0 Conclusions and recommendations

The Terminal Classic Period (800-1000 A.D.) was a very interesting period climatologically on the Yucatan Peninsula. The droughts that are said to have plagued the Yucatan Peninsula during the TCP may have been a driving factor in the collapse of the Maya Civilization, and their existence suggests that droughts of this magnitude could occur again. The goal of this study was to analyze multiple proxy records from the Yucatan Peninsula using a changepoint analysis to provide new insight into how climate conditions may have varied spatially across the Yucatan Peninsula during this time period. Collectively, the 23 proxy records analyzed do not show an overwhelming evidence of drought across the Yucatan Peninsula during; however, subdividing the records into regions on the Yucatan Peninsula shows that there was spatial variability in the occurrence of droughts during the TCP. The strongest evidence of TCP droughts is found in the records from Lake Chichancanab, Punta Laguna, and Tecoh Cave, while there is inconclusive evidence of drought in the records from Barranca de Amealco (central Mexico), Macal Chasm (Belize), Yok Balum Cave (Belize), and Lake Petén Itzá (Guatemala). Aguada X'Caamal, which is located on northwest Yucatan Peninsula, shows limited evidence of droughts, despite being located close to the three sites that do show evidence of droughts. As all of the proxy records from this site were oxygen isotope proxies (multiple proxy types were not available for this site), it is possible that this result is due to uncertainties inherent in the proxy records. Evidence from Yok Balum Cave and Macal Chasm suggest the short episodes of intense droughts occurred after the TCP in the southern Yucatan Peninsula, while the records from Lake Punta Laguna suggest that the onset of TCP droughts was earlier in the northwestern Yucatan Peninsula than in the other northern regions.

Analysis of three modes of variability which are known to affect the movement of the ITCZ (ENSO, PDO, and AMO) showed that for the first half of the TCP, all of these climate indices were in positive phase, which may have contributed to enhanced aridity during the Yucatan Peninsula dry season due to a southward shift of the ITCZ. However, all of these records were already, on average, in a positive phase for nearly 500 years before the TCP, suggesting their combined effects on the migration of the ITCZ were not the only cause of the TCP droughts.

While this study shows that ITCZ movement was not the sole contributor to drought variability on the Yucatan Peninsula during the TCP, there is still much work to be done to fully understand the mechanisms that may have caused these droughts. The following are recommendations for future work in this area:

1. **Collect more proxy records on the Yucatan Peninsula**. This would not only assist with corroborating the findings of the existing proxy records, but collecting more proxy records of different types would be much more useful in a multi-proxy approach. In addition, the priority should be placed on finding proxy records that record annual or seasonal changes (in precipitation, E/P, or another component of the hydrologic cycle) to provide more detailed drought identification during the TCP and to identify which season these droughts occurred in and how they may be related to the ITCZ, as well modes of variability which affect the ITCZ movement (such as AMO, PDO and ENSO).

2. **Comparison of oxygen isotope proxy records to an isotope-enabled General Circulation Model (GCM)**. Another method of studying the paleoclimate is to use paleoclimate simulations from GCMs as a



way to compare and validate the proxy records. The use of oxygen isotope type proxies in particular has
many uncertainties; therefore, output from an isotope enabled GCM could be compared to the oxygen
isotope type proxies to help better understand which atmospheric processes are affecting these proxies on
the Yucatan Peninsula.
3. **More study is needed to determine the mechanisms causing droughts**. This study showed that the
patterns of drought variability on the Yucatan Peninsula were complex, and this just highlights the need to
further understand the mechanisms that created them. While this study suggests that the ITCZ was not the
only cause, it would be useful to look at the effects of other potential mechanisms in addition to the effects
of the ITCZ position, as it likely played a role.
**Data availability**
All proxy data used for this analysis are available for download from the National Oceanic and Atmospheric
Administration (NOAA) National Centers for Environmental Information website. The paleoclimate dataset
webpage can be found at: https://www.ncdc.noaa.gov/data-access/paleoclimatology-data/datasets.
**Funding**
This research was supported by a Natural Sciences and Engineering Research Council (NSERC) Discovery Grant to
Diana Allen.
**Author contributions**
S.H. and D.M.A. conceived the study. The analyses were carried out by S.H. under the supervision of D.M.A. and
K.E.K.
**Declarations of Interest**
None.

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



**Figure 1. A-H.  Time series of proxy records used in this study from 380 A.D. to the present.  δ¹⁸O shell records are indicated by a shell, sediment-related proxies with yellow dots, δ¹⁸O of calcite encrusted algae with seaweed, tree rings with a tree, and speleothem with an inverted triangle.  The arrow on the right side shows the tendency for a record to indicate dry climate.  Time periods discussed in the text as evidence of drought are indicated with small purple arrows along the time series.**



755

**Figure 1 Continued. I-Q.** Time series of proxy records used in this study from 380 A.D. to the present. δ¹⁸O shell records are indicated by a shell, sediment-related proxies with yellow dots, δ¹⁸O of calcite encrusted algae with seaweed, tree rings with a tree, and speleothem with an inverted triangle. The arrow on the right side shows the tendency for a record to indicate dry climate. Time periods discussed in the text as evidence of drought are indicated with small purple arrows along the time series.




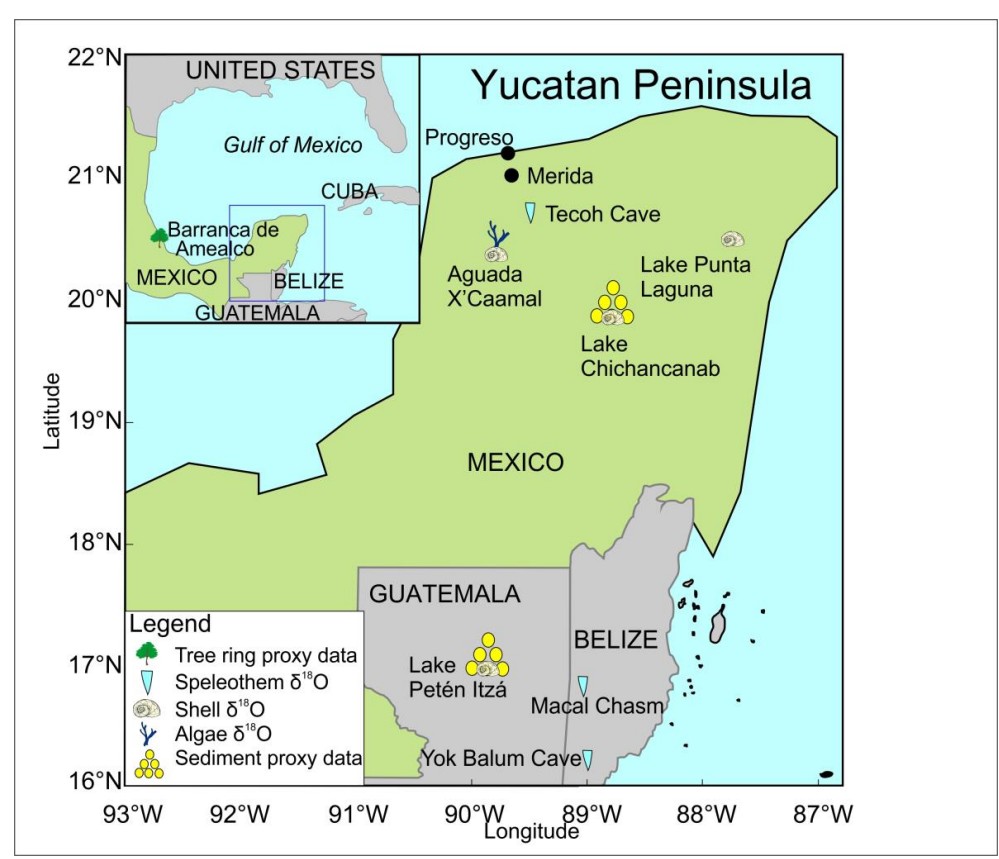

**Figure 2. Proxy data sites from the Yucatan Peninsula. Multiple proxy data types were found for three of the sites (see Supplementary Information S1). The location of the tree ring record (Barranca de Amealco) is shown in the inset map.**






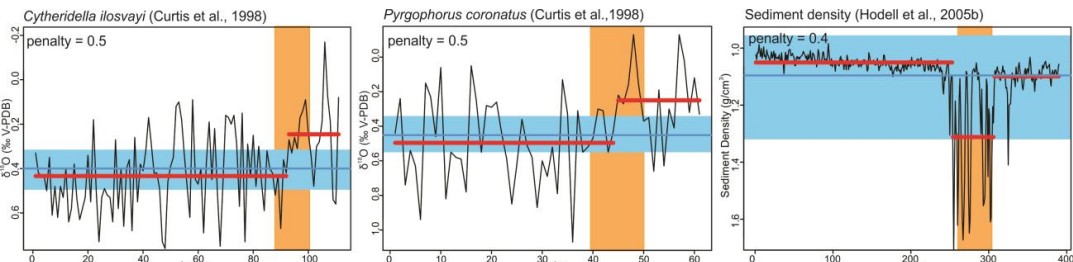


**Figure 3(a). Example results graphs from the changepoint analysis (mean). In each graph, the orange shaded area indicates the data points that fall within the TCP. Changepoints in mean are identified with breaks in the horizontal red lines which indicate the mean value of each segment in the time series. Penalty values are indicated in the top left corner in each graph; where the penalty value is not stated, the default penalty (log(n)) was found to be suitable for that record.**

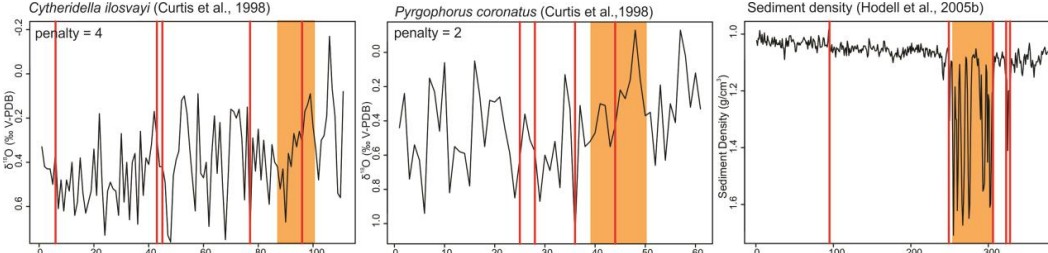

771

**Figure 3(b). Example results graphs from the changepoint analysis for mean (top) and variance (bottom). In each graph, the orange shaded area indicates the data points that fall within the TCP, and the blue shaded area indicates a +/- 20% change in the mean. Changepoints in variance are identified with vertical red lines. Penalty values are indicated in the top left corner in each graph; where the penalty value is not stated, the default penalty (log(n)) was found to be suitable for that record.**



| Site | Type | Proxy name | Region | Source | # Age Dates | Confidence | |
|------|------|-----------|--------|--------|-------------|-----------|---|
| Aguada X'Caamal | Shell | *D. stevensoni* δ¹⁸O (2) | Northwest | 1 | 4 | Moderate | ✓ |
| Aguada X'Caamal | Shell | *D. stevensoni* δ¹⁸O (3) | Northwest | 1 | 4 | Moderate | ✓ |
| Aguada X'Caamal | Shell | *D. stevensoni* δ¹⁸O (4) | Northwest | 1 | 4 | Moderate | ✓ |
| **Aguada X'Caamal** | **Algae** | **Chara δ¹⁸O (2)** | **Northwest** | **1** | **6** | **Moderate** | ✓ |
| **Aguada X'Caamal** | **Algae** | **Chara δ¹⁸O (3)** | **Northwest** | **1** | **6** | **Moderate** | ✓ |
| **Aguada X'Caamal** | **Algae** | **Chara δ¹⁸O (4)** | **Northwest** | **1** | **6** | **Moderate** | ✓ |
| **Aguada X'Caamal** | **Shell** | ***P. coronatus* δ¹⁸O (2)** | **Northwest** | **1** | **8** | **Moderate** | |
| Aguada X'Caamal | Shell | *P. coronatus* δ¹⁸O (3) | Northwest | 1 | 8 | Moderate | ✓ |
| Aguada X'Caamal | Shell | *P. coronatus* δ¹⁸O (4) | Northwest | 1 | 8 | Moderate | ✓ |
| ★ **Tecoh Cave** | **Speleothem** | **Chaac δ¹⁸O** | **Northwest** | **2** | **11** | **High** | ✓ |
| Barranca de Amealco | Tree ring | PDSI reconstruction | West | 3 | Crossdating | High | |
| ★ **Chichancanab** | **Shell** | ***Cyprinotus sp.* δ¹⁸O** | **Central** | **4** | **1** | **Low** | ✓ |
| Chichancanab | Shell | *Pyrgophorus sp.* δ¹⁸O | Central | 4 | 1 | Low | |
| **Chichancanab** | **Shell** | ***Physocypria sp*. δ¹⁸O** | **Central** | **4** | **1** | **Low** | ✓ |
| Chichancanab | Sediment | Calcite (%CaCO₃) | Central | 4 | 1 | Low | |
| ★ **Chichancanab** | **Sediment** | **Sediment density** | **Central** | **5** | **9** | **Moderate** | ✓ |
| ★ **Punta Laguna** | **Shell** | ***Pyrgophorus coronatus* δ¹⁸O** | **Northeast** | **6** | **4** | **Moderate** | ✓ |
| ★ **Punta Laguna** | **Shell** | ***Cytheridella ilosvayi* δ¹⁸O** | **Northeast** | **6** | **4** | **Moderate** | ✓ |
| Petén-Itzá | Shell | *Cytheridella ilosvayi* δ¹⁸O | South | 7 | 5 | Moderate | ✓ |
| Petén-Itzá | Shell | *Pyrgophorus coronatus* δ¹⁸O | South | 7 | 5 | Moderate | ✓ |
| Petén-Itzá | Sediment | Magentic susceptibility | South | 8 | 1 | Low | |
| **Macal Chasm** | **Speleothem** | **Macal δ¹⁸O** | **South** | **9** | **5** | **Moderate** | ✓ |
| Yok Balum Cave | Speleothem | Yok Balum δ¹⁸O | South | 10 | 34 | High | ✓ |

**Figure 4. Result of the changepoint analysis of 23 proxy records for the Yucatan Peninsula. Proxy data are separated in five regions (Northeast, Central, Northwest, West, and South) to aid in spatial interpretation. See Table S1 for additional details for each record. Numbers in parentheses indicate different sample numbers of species of the same name at the same site. Records that had a mean changepoint during the TCP are shaded in red (change towards drier conditions) or blue (change towards wetter conditions); darker shading indicates that the change was greater than a 20% difference from the mean of the entire time series. Records in bold are those which had a TCP mean that is significantly different from the time series mean (determined from the t-test). Green checkmarks are placed to the right of records which have a variance changepoint during the TCP. The five records that met all four inclusion criteria are indicated with a star. Proxy data sources: (1) Hodell et al., 2005a (2) Medina-Elizalde et al., 2010 (3) Stahle et al., 2011 (4) Hodell et al., 1995 (5) Hodell et al., 2005b (6) Curtis and Hodell, 1996 (7) Curtis et al., 1998 (8) Escobar, 2010 (9) Akers et al., 2016 (10) Kennett et al., 2012.**





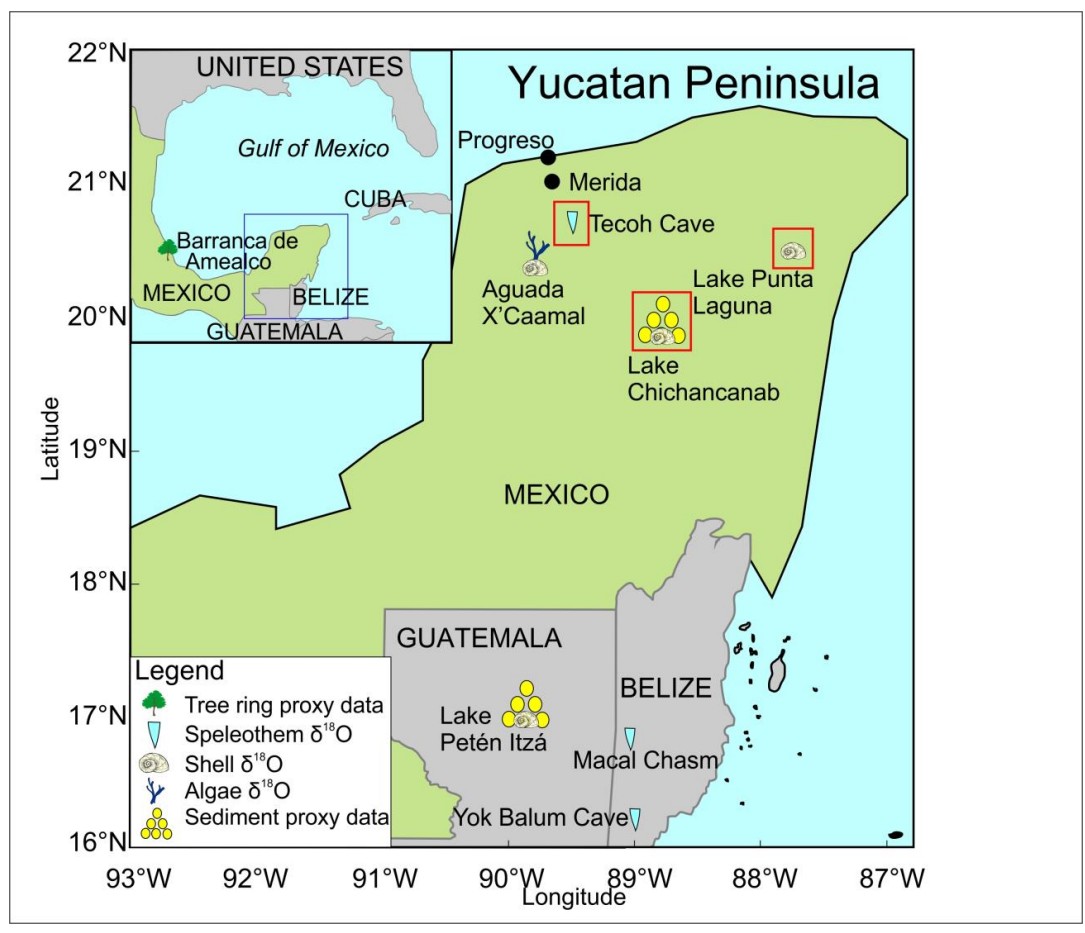

Figure 5. The two locations on the Yucatan Peninsula which had records meeting all four of the inclusion criteria for drought (highlighted in red boxes).

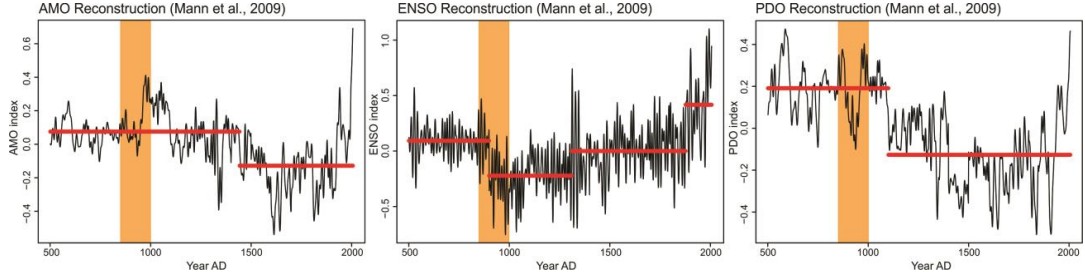

Figure 6. Mean changepoint analysis results for reconstructions of AMO, ENSO, and PDO from Mann et al. (2009). Red lines indicate the mean index value at for each segment of the record, with a changepoint represented as a break in the mean value. The TCP (850-1000 A.D.) is highlighted in orange.