# Peer review of "Spatial and temporal variability of Terminal Classic Period droughts from multiple proxy records on the Yucatan Peninsula, Mexico"

_Climate of the Past, 2019_

## Short Comment (SC1) · 28 Jun 2019

Hunter et al. compile previously published proxy records and perform statistical tests (namely changepoint analysis) to provide evidence for drying in the Yucatán Peninsula during the TCP.

In short, the manuscript provides a background discussion of (i) previously proposed causal forcing mechanisms that may have triggered the TCP droughts, and (ii) the previously published proxy records and their limitations. Subsequently, the manuscript goes on to discuss (i) the link between migration of the ITCZ and TCP droughts, (ii) the spatial and temporal variability of the droughts.

The results produced by Hunter et al. show that there are key differences in the spacial and temporal drying signatures in the proxy records. Comparison of the data to changepoint analysis conducted on PDO, ENSO and AMO signals led the authors to conclude that the position of the ITCZ was not the sole driver of the TCP droughts.

Whereas the idea of a coherent interval of drought across Mesoamerica has been explored in previous studies, the authors describe this manuscript as "novel" in its attempts to "(a) minimize uncertainties inherent to individual proxy records, (b) identify any regional patterns, and (c) explore temporal shifts in drought occurrence during the TCP". However, a large proportion of the theory and analysis presented in the manuscript has been extensively discussed in previously published literature:

1. Bhattacharya et al. (2017) synthesise regional proxy records to suggest that there is statistically robust evidence of drying between 800 and 1050 CE. By studying control simulations of two general circulation models (GCMs) and the major modes of climate variability (e.g. ENSO, NAO, etc), the authors also suggest that the pattern of drying may be diagnostic of hydroclimate changes associated with multidecadal variability in the Atlantic Basin. Overall, both the proxy analysis techniques and discussion points are very similar to those utilised by Hunter et al.. Critically, however, Bhattacharya et al. (2017) provide a much more in-depth review of literature proxy age models (using Bayesian age modelling) and causal drought mechanisms (specifically by using GCM control simulations to analyse the processes that produce long-term droughts, including PDO, ENSO and AMO signals). Note that this paper is not cited in the text.

2. Douglas et al. (2016a) provide an in depth review of previous proxy records, and perform z score analysis to highlight any deviation of individual proxy measurements from the mean value of a record. Douglas et al. (2016a) observe two distinct dry intervals, at 770–950 and 1000–1100 CE, separated by an intervening period of relatively wet conditions, as well as spacial variability between records (also see Douglas et al., 2016b).
3. Evans et al. (2018) also use Bayesian age modelling and the 'Events' function in R to quantify drought timing in multiple proxy records, and provide a robust method to quantify the drought at Lake Chichancanab, one of the sites highlighted by the authors. This paper is also not cited in the text.

Could the authors elaborate as to how their manuscript provides a significant advance to the papers and analysis listed above?

Points that would significantly enhance the manuscript:

1. The use of published age models and the assumption they are accurate (which is highly unlikely) is critical to the subsequent comparison of the data to changepoint analysis conducted on PDO, ENSO and AMO signals. As a bare minimum, Bayesian age analysis should be used to quantify the errors in the age models. Sites with <5 radiocarbon (or other) dates in the last 2000-year interval should not be used (see Bhattacharya et al., 2017).

2. Each data set used by Hunter et al. spans a different range of calendar dates; thus, the mean values for each data set derive from different time intervals and may slightly bias interpretations of drought intensity inferred from the changepoint analysis in each record. This should be discussed in addition to Section 4.3, as well as in relation to the 20% uncertainty associated to the changepoint analysis.

3. Note that records of gypsum or calcium carbonate concentrations in sediments should be excluded from changepoint analysis: because mineral precipitation occurs at a chemical threshold (on/off) related to the saturation state of gypsum or carbonate minerals, these records are likely not linearly related to the magnitude of reductions in rainfall. Also, these records do not record wet climate intervals, as periods of wetter than average climate do not cause changes in mineral precipitation (Douglas et al., 2016). Additionally, tree rings records should also be excluded as existing records from Mesoamerica generally reflect early spring rainfall, and may reflect a distinct climatic signal from that recorded by speleothem and lake records (Bhattacharya et al., 2017).

4. The lack of statistical linkage between the analysis of (i) the proxy datasets and (ii) the modes of climate variability means that the assessment is only qualitative. After the points above have been considered, statistical test should be used to quantify the likelihood that PDO, ENSO and AMO signals are forcing drought conditions (see see Bhattacharya et al. [2017] for further discussion of mechanistic linkages).

5. A full discussion as to why there is a discrepancy between the Chaac and Yok I speleothems would provide useful insight (building on the explanation of Douglas et al., 2016b), as well as comparison of the statistical techniques used in this manuscript and the paper of Bhattacharya et al. (2017).

6. Section 4.3 adds little to the discussion, especially given proxy "uncertainty" is discussed earlier in the text. Indeed, the importance of the proxy dating errors (see above) means that the discussion in this section should be considered prior to statistical analysis.

Other points to note:

1. Lines 170-184: Please see the paper by Evans et al. (2018) who used triple oxygen and hydrogen isotopes of gypsum hydration water to provide the a robust, quantitive estimate of precipitation changes during the TCP and deconvolve relative humidity and rainfall.

2. Lines 185-191: Note that lake level records are sensitive to E-P. In contrast, speleothem records are generally interpreted in terms of the 'amount effect' with isotope ratios covarying with the amount of annual precipitation.

3. Lines 201-203. The largest uncertainty with the record of Medina-Elizalde et al. (2010) is the relatively poor correlation (low $r^2$ value) in the modern calibration period, as well as the fact that the magnitude of $\delta 18O$ variability in the Chaac record spans a much wider range than the magnitude of $\delta 18O$ variability from the recent calibration period (see Douglas et al., 2016a).

4. A clearer structure in the results section (e.g. segment the proxies by geography) would help the reader.

5. The use of qualitative language (e.g. line 364: "a bit less certain"; line 392: "more certain" etc) should be replaced by quantitative results.

References:

T. Bhattacharya, J. C. H. Chiang, W. Cheng, Ocean-atmosphere dynamics linked to 800-1050 CE drying in Mesoamerica. Quat. Sci. Rev. 169, 263–277 (2017).

P. M. Douglas, A. A. Demarest, M. Brenner, M. A. Canuto, Impacts of climate change on the collapse of lowland Maya civilization. Annu. Rev. Earth Planet. Sci. 44, 613–645 (2016a).

P. M. Douglas, M. Brenner, J. H. Curtis, Methods and future directions for paleoclimatology in the Maya lowlands. Global Planet. Change 138, 3–24 (2016b).

N. P. Evans, T. K. Bauska, F. Gázquez-Sánchez, M. Brenner, J. H. Curtis, D. A. Hodell, Quantification of drought during the collapse of the classic Maya civilization. Science. 361, 6401, 498-501 (2018).

---

## Author Comment (AC1) · 8 Jul 2019

Comments: Hunter et al. compile previously published proxy records and perform statistical tests (namely changepoint analysis) to provide evidence for drying in the Yucatán Peninsula during the TCP. In short, the manuscript provides a background discussion of (i) previously proposed causal forcing mechanisms that may have triggered the TCP droughts, and (ii) the previously published proxy records and their limitations. Subsequently, the manuscript goes on to discuss (i) the link between migration of the ITCZ and TCP droughts, (ii) the spatial and temporal variability of the droughts. The results produced by Hunter et al. show that there are key differences in the special and

temporal drying signatures in the proxy records. Comparison of the data to changepoint analysis conducted on PDO, ENSO and AMO signals led the authors to conclude that the position of the ITCZ was not the sole driver of the TCP droughts. Whereas the idea of a coherent interval of drought across Mesoamerica has been explored in previous studies, the authors describe this manuscript as "novel" in its attempts to "(a) minimize uncertainties inherent to individual proxy records, (b) identify any regional patterns, and (c) explore temporal shifts in drought occurrence during the TCP". However, a large proportion of the theory and analysis presented in the manuscript has been extensively discussed in previously published literature:

Response: First, we would like to thank N. Evans for his very useful comments, and for taking the time to read and comment on our manuscript. In particular, we appreciate the new references that support our findings. Below we address each comment, and outline how we would revise our manuscript to address the concerns raised.

The main concern that Evans raises is that the data analysed in this study have been analysed previously by other authors. Evans questions whether our study is novel, and recommends that we expand on the new contributions made in this study. Indeed, the Terminal Classic Period droughts have been studied by many authors, including those whose proxy records we have analysed in this study. We believe that our research adds new insight into the complexity of the TCP droughts, and brings more detail to the idea of spatial variability in droughts across the Yucatán Peninsula by separating the proxy records into 5 regions (Northwest, West, Central, Northeast, and South). The changepoint analysis used in this study is a novel approach for analyzing proxy records on the Yucatan Peninsula; to our knowledge it has not been used in other proxy studies with multiple types of proxies, although it was used for the analysis of multiple sediment cores in one recent study (Trauth et al., 2018). This method has been used in numerous studies looking at climate and hydroclimate timeseries (references in lines 147-153 of the manuscript), and we wanted to apply this technique to a proxy analysis with multiple types of proxies, which can be difficult to analyse together due to the

different mechanisms which affect their formation. Our goal in using this statistical technique was to semi-quantitatively assess if the drought events that occurred during the TCP were in some way different from other dry periods that have been recorded in the proxy records. In our revisions, we will expand upon the motivation behind this study to make these goals more clear.

Three points were raised regarding previously published literature that we will address individually:

1. Comments: Bhattacharya et al. (2017) synthesise regional proxy records to suggest that there is statistically robust evidence of drying between 800 and 1050 CE. By studying control simulations of two general circulation models (GCMs) and the major modes of climate variability (e.g. ENSO, NAO, etc.), the authors also suggest that the pattern of drying may be diagnostic of hydroclimate changes associated with multidecadal variability in the Atlantic Basin. Overall, both the proxy analysis techniques and discussion points are very similar to those utilised by Hunter et al.. Critically, however, Bhattacharya et al. (2017) provide a much more in-depth review of literature proxy age models (using Bayesian age modelling) and causal drought mechanisms (specifically by using GCM control simulations to analyse the processes that produce long-term droughts, including PDO, ENSO and AMO signals). Note that this paper is not cited in the text.

Response: Bhattacharya et al. (2017) synthesized regional proxy records from the Yucatán Peninsula and found evidence of drying between 800 and 1050 CE. The authors suggest that drying is associated with Atlantic multidecadal variability. Bhattacharya et al. (2017) touch on the idea of spatial and temporal variability in the TCP droughts (by noting that some sites show a continuous dry interval from 800 to 1050 CE, while others show several dry intervals). Our study goes into more detail about the spatial variability of droughts on the Yucatán Peninsula by grouping the sites into geographic regions and providing a more in depth comparison between these regions. We also offer a new statistical method for analysing the proxy data, the changepoint analysis,

which helps to identify periods in each time series that have are statistically different conditions compared to the rest of the time series. It should also be noted that our paper focuses solely on drought during the Terminal Classic Period (TCP) rather than the years 0 CE to present as in Bhattacharya et al. (2017). In revising our manuscript, we will include some discussion points related to this paper, and elaborate on how it differs from our study.

2. Comments: Douglas et al. (2016a) provide an in depth review of previous proxy records, and perform z score analysis to highlight any deviation of individual proxy measurements from the mean value of a record. Douglas et al. (2016a) observe two distinct dry intervals, at 770–950 and 1000–1100 CE, separated by an intervening period of relatively wet conditions, as well as spatial variability between records (also see Douglas et al., 2016b).

Response: Indeed, Douglas et al. (2016a) provide an in depth review of many of the same proxy records analysed in this study. A review of the proxy records is included in our manuscript as well, as the mechanisms which allow relation of these proxies to drought conditions are important for the interpretation of these proxy records. The z-score used in Douglas et al. (2016a) looks at deviation of each data point from the mean of the proxy time series, and normalizes all the studied proxy records. While the resampling procedure used helps to estimate drought intensity, we argue that the identification of drought in Douglas et al. (2016a) is still somewhat qualitative- the z-score time series are visually analyzed to identify the periods of drought. As there are natural variations between wet and dry periods in all of the records studied, the intention of our paper is to explore the changepoint method as a means to semi-quantitatively assess which periods of drought were particularly different from other periods. Specifically, were the TCP droughts statistically different from other droughts in the area? We could expand on this in the introduction section of this paper, to help explain our motivation for trying this changepoint methodology on proxy records which have previously been studied. In regards to the spatial variability of droughts, the variability

discussed in Douglas et al. (2016a, 2016b) focuses on the difference between the north and south Yucatán, but not the differences within the northern and/or southern regions. This is seen in the division of the z-score analysis into the Northern Yucatán Peninsula and Belize/Central Petén, the comparison of $\delta$Dwax records from Lake Chichancanab (northern Peninsula) and Lake Salpetén (southern Peninsula), and the comparison between the Chaac (northern Peninsula) and Yok Balum (southern Peninsula) speleothem records. Our study looks at spatial variability in finer detail by splitting the Yucatán Proxy records into Northwest, West, Central, Northeast, and South subgroups. A subtle difference, but an important one because the differences found in our study show that the spatial variability of these droughts was more complex that just a difference between north and south- there were variations from east to west as well, which suggests that local factors may have contributed to the onset of drought in certain areas, while other areas may have experienced no droughts or smaller magnitude droughts. This also supports the conclusion of the manuscript that north-south shifts in the position of the ITCZ weren't the only control on the occurrence of droughts.

3. Comments: Evans et al. (2018) also use Bayesian age modelling and the 'Events' function in R to quantify drought timing in multiple proxy records, and provide a robust method to quantify the drought at Lake Chichancanab, one of the sites highlighted by the authors. This paper is also not cited in the text. Could the authors elaborate as to how their manuscript provides a significant advance to the papers and analysis listed above?

Response: We acknowledge that quantification of drought using the available proxy records is very difficult given the nature of the various proxy types and their relationships to evaporation and precipitation, which is why we chose to take a semi-quantitative approach to this study. This recent paper by Evans et al. (2018) provides an interesting multi-proxy approach to quantifying precipitation at Lake Chichancanab, and we would include a discussion of this paper in the introduction of our manuscript alongside our discussion of Medina-Elizalde et al. (2010), which also attempted to

quantify precipitation during the TCP using speleothem $\delta$18O.

We believe we have discussed in the responses above how our study advances the narrative of TCP droughts; overall, it aims to provide a way to statistically compare the TCP droughts to the climate variability observed in the entire length of the proxy records (the changepoint analysis). It also uses the changepoint analysis to determine if PDO, ENSO, and AMO (all related to the position of the ITCZ) were significantly different at the time of the TCP droughts. Finally, the discussion of spatial variability also notes differences in the timing of droughts from west to east across the Yucatán Peninsula, whereas previous papers have focuses only on the north to south differences.

Comments: Points that would significantly enhance the manuscript:

1. Comments: The use of published age models and the assumption they are accurate (which is highly unlikely) is critical to the subsequent comparison of the data to changepoint analysis conducted on PDO, ENSO and AMO signals. As a bare minimum, Bayesian age analysis should be used to quantify the errors in the age models. Sites with <5 radiocarbon (or other) dates in the last 2000-year interval should not be used (see Bhattacharya et al., 2017).

Response: Upon revising this paper, we could expand on our assessment of the age models by employing Bayesian age analysis to quantify age uncertainty. We agree that sites with fewer than five age dates have much more uncertainty than those with more age dates. We attempted to account for this by ranking each proxy record in terms of confidence in that record. We don't believe it is necessary to completely exclude them from the analysis, as the point of this paper was to look at the available proxy records for the Yucatán Peninsula as a whole. We could, however, make it more clear in our conclusions that these records should not be considered to have definitive evidence of drought.

2. Comments: Each data set used by Hunter et al. spans a different range of calendar dates; thus, the mean values for each data set derive from different time intervals

and may slightly bias interpretations of drought intensity inferred from the changepoint analysis in each record. This should be discussed in addition to Section 4.3, as well as in relation to the 20% uncertainty associated to the changepoint analysis.

Response: We agree. This can be added as part of the discussion about uncertainty in the proxy records, and in particular the ones that have fewer data points during the TCP.

3. Comments: Note that records of gypsum or calcium carbonate concentrations in sediments should be excluded from changepoint analysis: because mineral precipitation occurs at a chemical threshold (on/off) related to the saturation state of gypsum or carbonate minerals, these records are likely not linearly related to the magnitude of reductions in rainfall. Also, these records do not record wet climate intervals, as periods of wetter than average climate do not cause changes in mineral precipitation (Douglas et al., 2016). Additionally, tree rings records should also be excluded as existing records from Mesoamerica generally reflect early spring rainfall, and may reflect a distinct climatic signal from that recorded by speleothem and lake records (Bhattacharya et al., 2017).

Response: The calcium carbonate and magnetic susceptibility (which are both related to gypsum content) records have both been previously published as recorders of Mesoamerican drought (Hodell et al., 1995; Escobar, 2010). They are likely not related to rainfall, but to the ratio of evaporation and precipitation (E/P). Douglas et al. (2016b) also discusses that gypsum and magnetic susceptibility have been used to identify periods of droughts. We have included these records due to their association with E/P and ability to record arid events, not because they are correlated to rainfall or to record wet events. The tree ring record used in this study is a reconstruction of June PDSI (not spring rainfall), and is also not directly located on the Yucatán Peninsula, but to the west of it. While the Barranca de Amealco tree ring record is strongly correlated to April-June precipitation, it was also strongly correlated with June PDSI, and therefore was used as an indicator of Mesoamerican drought. It was included in this

analysis for the purpose of comparison, but as discussed in lines 414-418, this record did not show evidence of TCP droughts (nor was it expected to, which helps to validate the changepoint method used). A discussion of the interpretation of tree ring records in Mesoamerican could be added in revisions to this manuscript, to highlight the fact that they tend to respond to difference climate signals than other proxy types.

4. Comments: The lack of statistical linkage between the analysis of (i) the proxy datasets and (ii) the modes of climate variability means that the assessment is only qualitative. After the points above have been considered, statistical test should be used to quantify the likelihood that PDO, ENSO and AMO signals are forcing drought conditions (see Bhattacharya et al. [2017] for further discussion of mechanistic linkages).

Response: We agree, this analysis would benefit by statistically relating the climate teleconnection signals to the proxy records, and we would add this to our future revision.

5. Comment: A full discussion as to why there is a discrepancy between the Chaac and Yok I speleothems would provide useful insight (building on the explanation of Douglas et al., 2016b), as well as comparison of the statistical techniques used in this manuscript and the paper of Bhattacharya et al. (2017).

Response: We can expand on the discussion of the discrepancies between the Chaac and Yok I speleothems in our revisions. Based on the regional patterns observed in the proxy records, we believe that the discrepancies are recording actual differences in drought timing between these two regions, as 3 out of 5 of the records in the south region appear to have a wet climate during the TCP. This is in contrast to Douglas et al., (2016b), who suggest these differences may be due to evaporation and kinetic isotope effects. We can also include a section outlining how the statistical techniques used in this paper are different from Bhattacharya et al. (2017).

6. Comment: Section 4.3 adds little to the discussion, especially given proxy "uncertainty" is discussed earlier in the text. Indeed, the importance of the proxy dating errors (see above) means that the discussion in this section should be considered prior to statistical analysis.

Response: We agree that perhaps section 4.3 is redundant- we suggest taking important points from Section 4.3 and incorporating them into our earlier discussion of proxy uncertainty, so that all discussion of the uncertainties occur prior to the changepoint analysis.

Comment: Other points to note:

1. Comment: Lines 170-184: Please see the paper by Evans et al. (2018) who used triple oxygen and hydrogen isotopes of gypsum hydration water to provide the a robust, quantitive estimate of precipitation changes during the TCP and deconvolve relative humidity and rainfall.

Response: We note that Evans et al. (2018) conducted work that is relevant to this section, and can add a discussion of these points of the paper to this section.

2. Comment: Lines 185-191: Note that lake level records are sensitive to E-P. In contrast, speleothem records are generally interpreted in terms of the 'amount effect' with isotope ratios covarying with the amount of annual precipitation.

Response: We agree. Lines 185-190 are not discussing speleothem $\delta18O$, but shell $\delta18O$ of lake fossils, and this can be clarified. Lines 191-209 discuss speleothem records. We mention the correlation of the Chaac speleothem with precipitation here, as this is the reasoning given in Medina-Elizalde et al. (2010) for interpreting the speleothem as recording variations in precipitation. We can add an explanation of the amount effect here as well to further explain how lake level and speleothem records differ in their interpretation.

3. Comment: Lines 201-203. The largest uncertainty with the record of Medina-Elizalde et al. (2010) is the relatively poor correlation (low rË̤2 value) in the modern calibration period, as well as the fact that the magnitude of _18O variability in the Chaac record spans a much wider range than the magnitude of _18O variability from the recent calibration period (see Douglas et al., 2016a).

Response: Lines 201-203 are addressing the assumption of stable temperatures during $\delta18O$ proxy formation. However, you make a good point. Perhaps this point is best made in the assessment of individual proxy record uncertainty (the current Section 4.3, but which would be moved before the methods as per your previous suggestion).

4. Comment: A clearer structure in the results section (e.g. segment the proxies by geography) would help the reader.

Response: We organized the results section by site, but we could reorganize and clarify this using subheaders for each of the geographical regions analysed.

5. Comment: The use of qualitative language (e.g. line 364: "a bit less certain"; line 392: "more certain" etc) should be replaced by quantitative results.

Response: We agree. We believe this could be improved upon by including the Bayesian age analysis to help quantify uncertainty in the age models.

References: T. Bhattacharya, J. C. H. Chiang, W. Cheng, Ocean-atmosphere dynamics linked to 800-1050 CE drying in Mesoamerica. Quat. Sci. Rev. 169, 263–277 (2017). P. M. Douglas, A. A. Demarest, M. Brenner, M. A. Canuto, Impacts of climate change on the collapse of lowland Maya civilization. Annu. Rev. Earth Planet. Sci. 44, 613–645 (2016a). P. M. Douglas, M. Brenner, J. H. Curtis, Methods and future directions for paleoclimatology in the Maya lowlands. Global Planet. Change 138, 3–24 (2016b). N. P. Evans, T. K. Bauska, F. Gázquez-Sánchez, M. Brenner, J. H. Curtis, D. A. Hodell, Quantification of drought during the collapse of the classic Maya civilization. Science. 361, 6401, 498-501 (2018). Trauth, M.H., Foerster, V., Junginger, A., Asrat, A., Lamb, H.F., and Schaebitz, F.: Abrupt or gradual? Change 746 point analysis of the late Pleistocene- Holocene climate record from Chew Bahir, Southern Ethiopia, Quaternary

747 Research, 90, 321-330, 2018.

---

## Referee Comment (RC1) · Matthew Peros (Referee) · 17 Sep 2019

General comments: Thank you for this opportunity to review the manuscript "Spatial and temporal variability of Terminal Classic Period droughts from multiple proxy records on the Yucatan Peninsula, Mexico", by Stephanie Hunter, Diana Allen, and Karen Kohfeld. The manuscript seeks to: 1) objectively and systematically identify drought events in a number of Yucatan proxy indicators and determine to what extent these correspond to the Terminal Classic Period (TCP); 2) identify spatial and temporal differences among these records, and 3) assess potential driving mechanisms of drought events. Some of the manuscript's positive points are its discussion of limitations in

the data and the application of an apparently objective set of criteria to identify hydro-climatic changes in the proxy records. Overall I think this is an important paper with relevance for multiple fields (paleoclimatology/archaeology) and that it is publishable in Climate of the Past following revisions, and I find the manuscript takes into account (or can be improved so that it does) the criteria/aspects that are outlined under the review criteria on the Climate of the Past website.

Specific comments: Fundamentally, the manuscript relies on a comparison of presumed droughts based on the proxy data and its comparison to the TCP. But, unless I missed it, we are left to take the timing of the TCP at face value as 800-100 A.D. What is this date range based on? There are a number of citations in the first sentence (Lines 33-36) but these citations are essentially the proxy data that are used in this paper. In much the same way as the manuscript has a good discussion of the meaning and limitations of the proxy data, I think it would benefit from a short discussion of the actual TCP from an archaeological point of view. What archaeological data are used? What limitations are there in that data? I am not an expert on the archaeology of the region, but my understanding is that the "collapse" – or the period of time this transition occurred - was time transgressive (i.e., occurred at different times at different places). While it might not be in the scope of this paper to attempt to plot those vertical orange bars at different times based on location, acknowledging the nuance in the timing of the TCP that the proxy records are being compared to would be useful, in my opinion.

(And on a related note, is the TCP 800-1000 A.D. , or 850-1000 A.D.?) The caption and I think orange bars in Fig. 6 place it at 850-1000 A.D. whereas it is 800-100 AD elsewhere. And should dates be reported in C.E. and not A.D.?)

The use of changepoint analysis is interesting and a useful approach I think. I can see how it would be useful to identify changes in mean state (as in Fig 3a) but I wonder about its utility for assessing variance (Fig 3b). And my concern here is that within each timeseries (unless it is the tree ring data which I assume is annual), the temporal spacing (or timing) of adjacent proxy measurements will vary based on the initial

sampling resolution and sedimentation/growth rates. I could see this being less of an issue for determining mean state, as I said, but I wonder to what extent this affects the variance measures. In this technique, does the data need to be evenly spaced, and if it is not, what kind of effect does this have on the results?

A large portion of the discussion is devoted to the question of whether the droughts were caused by ITCZ migration, and to do this the authors looks for corresponding changes in reconstructions of ENSO, PDO, and AMO. But to answer this question, would it not be better just to compare the data to a reconstruction of ITCZ position, such as one published by Lechleitner et al., (2017) (and cited on page 14), or possibly the Ti record from the Cariaco Basin (Haug et al.?) This would seem to be a more direct way to address the question.

And I think the analysis of the Mann et al., 2009 reconstructions was a good approach. It is interesting though, because individual proxy records of some of these climate modes show results that seem to differ from the Mann et al., 2009 reconstruction. For example, the Laguna Pallcacocha, Ecuador data (Moy et al., 2002) seems to show positive (warm) phase ENSO between about 800-1100 AD, which would be consistent with southward displacement of ITCZ and drought on the Yucatan. Interestingly, I think (but I could be wrong) that the Mann et al., 2009 reconstruction is based in part on this dataset, but the point is that there is reliable proxy data (from individual sites) that records different activity than the large-scale reconstructions.

I understand why the charts in Fig 3a and b are plotted by "Index value", which I think is basically the number of the sample starting from the earliest one, but why are the reconstructions in Fig 6 plotted by age? I assume the Mann et al., 2009 reconstructions are annual (I haven't checked recently) but it seems inconsistent.

Finally, there are improvements that could be made to the figures/tables captions to make the manuscript easier to understand. For example, the caption for Figure 5 says that two locations had records that meet all 4 criteria, and that these are highlighted by

red boxes, but there are three red boxes at three locations.

Technical corrections: Line 115: "A couple of" is too casual – please reword. Line 722: There is something unclear about this figure caption. . . does the mean (top) need to be mentioned when there is already a caption for it? Line 778: Should this be a table and not a figure? Line 794: Typo "at for each"
* * *

---

## Referee Comment (RC2) · Anonymous Referee #2 · 27 Sep 2019

The problem that this paper tackles is an interesting one: there has been much debate about whether the so-called 'Terminal Classic' drought represents a coherent interval of climate change across the Yucatan Peninsula, and what dynamics may be responsible for the drought. This paper definitely has potential, but the authors should review additional relevant literature and reframe or expand some of their analyses.

First, why does the study only focus on the Yucatan Peninsula? If the authors are interested in looking for evidence of ITCZ changes, we would expect to see changes in the Caribbean, Gulf of Mexico, Central America, and northern South America. We may also expect to see antiphased changes in records from places farther south in

the Amazon. We also have records of tropical storms from the wider circum-Caribbean region (see the work of Jeff Donnelly's group at WHOI). Can we more rigorously test the idea that the ITCZ may have shifted in this interval by using additional proxy records? The claim of using of '23' proxy records is a little bit misleading because many of the proxy records are from the same sites, and therefore are not really independent datapoints.

Line 121: Given that pollen is not necessarily a linear indicator of forest cover, it is possible that there could have been intensified deforestation at the Terminal Classic – I recommend checking the land use reconstructions of Kaplan et al. 2011 "Anthropogenic Land Cover Change scenario for the preindustrial Holocene" to see what the reconstruction looks like in this particular region.

One factor that isn't really considered in this study is the timing of social change or site abandonment in the archaeological record – we know for a fact that this was not uniform across the Maya region - see for instance Aimers et al., 2007 – this could be discussed more in the paper.

The citations in this paper are not really up to date - A few other papers that already address some of the themes in this paper, in some cases with more detailed analyses of the climate dynamics and age models for each site, should be discussed and cited.

-Bhattacharya et al., 2017 in Quaternary Science Reviews includes a detailed analysis of the timing of drought in multiple records accounting for age uncertainty, and analyzes the drivers of drought in comprehensive climate models.

-Evans et al 2018 in Science used new measurements of gypsum hydration waters and lake level modeling to estimate large changes in precipitation at the Terminal Classic. The estimates stand in contrast to Medina-Elizalde and Rohling, 2012, which estimated a modest change in rainfall.

-There is also an interesting discussion in Metcalfe and Barron, 2015, which reviews

an extensive dataset of proxy records from across Mexico and parts of the Caribbean and Gulf of Mexico. These should be incorporated into the discussion, and can provide pointers on additional proxy records to incorporate into the text.

Line 430: El Nino events do increase winter rainfall in this region, but they actually decrease summer rainfall. This is because warm ENSO events generate an atmospheric Kelvin wave that dampens surface precipitation in much of the tropics – see Lintner et al., 2005, Journal of Climate. There is a delayed response in the following spring that enhances rainfall as a delayed response to ENSO.

Line 442: I am skeptical of the inferences of Knudsen et al about the inverse relationship of AMO precipitation and Yucatan rainfall – it runs counter to much of what we know about the dynamics of the region. See the work on the Atlantic Warm Pool by Wang et al., 2005 in Journal of Climate, as well as the work by Giannini et al. that is cited in this paper.

Overall, the paper addresses a topic worthy of study – it just needs revisions to the text and the inclusion of a greater number of proxy records to fully test the hypotheses it sets forth.

---

## Author Comment (AC2) · 2 Oct 2019

Comments: General comments: Thank you for this opportunity to review the manuscript "Spatial and temporal variability of Terminal Classic Period droughts from multiple proxy records on the Yucatan Peninsula, Mexico", by Stephanie Hunter, Diana Allen, and Karen Kohfeld. The manuscript seeks to: 1) objectively and systematically identify drought events in a number of Yucatan proxy indicators and determine to what extent these correspond to the Terminal Classic Period (TCP); 2) identify spatial and temporal differences among these records, and 3) assess potential driving mechanisms of drought events. Some of the manuscript's positive points are its discussion

of limitations in the data and the application of an apparently objective set of criteria to identify hydroclimatic changes in the proxy records. Overall I think this is an important paper with relevance for multiple fields (paleoclimatology/archaeology) and that it is publishable in Climate of the Past following revisions, and I find the manuscript takes into account (or can be improved so that it does) the criteria/aspects that are outlined under the review criteria on the Climate of the Past website.

Response: First, we would like to thank M. Peros for taking the time to review our manuscript, and for his insightful comments on our work. He has also helped us find some small typos that we missed, so we are very grateful for his attention to detail!

Comments: Specific comments: Fundamentally, the manuscript relies on a comparison of presumed droughts based on the proxy data and its comparison to the TCP. But, unless I missed it, we are left to take the timing of the TCP at face value as 800-100 A.D. What is this date range based on? There are a number of citations in the first sentence (Lines 33-36) but these citations are essentially the proxy data that are used in this paper. In much the same way as the manuscript has a good discussion of the meaning and limitations of the proxy data, I think it would benefit from a short discussion of the actual TCP from an archaeological point of view. What archaeological data are used? What limitations are there in that data? I am not an expert on the archaeology of the region, but my understanding is that the "collapse" – or the period of time this transition occurred - was time transgressive (i.e., occurred at different times at different places). While it might not be in the scope of this paper to attempt to plot those vertical orange bars at different times based on location, acknowledging the nuance in the timing of the TCP that the proxy records are being compared to would be useful, in my opinion.

Response: We noticed some differences across the literature in the time period chosen to represent the TCP; for example, Medina-Elizalde et al., (2010) takes the TCP to be the period from 800-950 C.E., while Turner and Sabloff (2012) says the TCP is from 800-1000 C.E., while Acuna-Soto et al. (2005) states the TCP is from 750-950 C.E. So while there are differences, we chose 800-1000 C.E. to be encompassing of the time

periods most commonly chosen to represent the TCP; however, we agree that adding a discussion of these differences and the archaeological evidence used to date the TCP would be an excellent addition to our manuscript, and can add this to the next version.

Comments: (And on a related note, is the TCP 800-1000 A.D. , or 850-1000 A.D.?) The caption and I think orange bars in Fig. 6 place it at 850-1000 A.D. whereas it is 800-100 AD elsewhere. And should dates be reported in C.E. and not A.D.?)

Response: Thank you for pointing this out- as noted above, there are some discrepancies in what the timing of the TCP is taken to be, but we have tried to be consistent throughout our manuscript. However, we made a mistake with Figure 6; the TCP should be labeled as 800-1000 A.D., and we will extend the shaded orange area to cover this time period as well (this does not affect the results of the analysis).

In addition, your comment about C.E. vs A.D. lead us to look into this more, as we were not aware of the connotations behind the two notations and believed it was simply a matter of preference (and noticed both used in the literature). We found the article published at https://www.thoughtco.com/when-to-use-ad-or-ce-116687 to be particularly interesting, and in the spirit of promoting inclusivity will be changing all the notation within our manuscript to C.E.

Comments: The use of changepoint analysis is interesting and a useful approach I think. I can see how it would be useful to identify changes in mean state (as in Fig 3a) but I wonder about its utility for assessing variance (Fig 3b). And my concern here is that within each timeseries (unless it is the tree ring data which I assume is annual), the temporal spacing (or timing) of adjacent proxy measurements will vary based on the initial sampling resolution and sedimentation/growth rates. I could see this being less of an issue for determining mean state, as I said, but I wonder to what extent this affects the variance measures. In this technique, does the data need to be evenly spaced, and if it is not, what kind of effect does this have on the results?

Response: We agree that in our analysis, the mean changepoints seem to be more

useful than then changes in variance. There were no consistent patterns observed in the variance changepoints across site locations or across the records we used; in fact, most of the records did show that there was a change in variance during or near the TCP, and it was rather the changes in the mean that helped to identify spatial patterns across proxy records. We wanted to include the variance changepoint analysis on the basis that, if more extreme droughts were observed during the TCP, perhaps this would show as a change in the variance at this time period. However, the results of the variance changepoint analysis was inconclusive in this regard. We think it would be useful to include a note about this in our discussion section in the next revision of our manuscript. In regards to your question about the spacing of points in the changepoint analysis, the concept of time is not necessarily taken into account; the r package (changepoint) reads in a list of ordered points that only have an index value associated with them. In the case of the proxy records, for those with lower than annual resolution, each point may represent anywhere from 2 years to a few decades (sample resolution is listed in our Supplementary Information, Table S1). Therefore, while it is possible to still conduct the variance analysis, the changepoints identified don't reflect a single year- they are more of a change in variance between the time periods represented by each measurement. This could be part of the reason why we were unable to identify a consistent pattern in the variance changepoints (that the low sample resolution was not able to capture rapid changes in variance that may have signified the TCP), or it's possible that similar changes in variance were observed all over the Yucatan Peninsula, and that it is not necessarily a useful criteria for identifying local droughts. These points will be added to our manuscript in our discussion of uncertainty regarding the changepoint analysis.

Comments: A large portion of the discussion is devoted to the question of whether the droughts were caused by ITCZ migration, and to do this the authors looks for corresponding changes in reconstructions of ENSO, PDO, and AMO. But to answer this question, would it not be better just to compare the data to a reconstruction of ITCZ position, such as one published by Lechleitner et al., (2017) (and cited on page 14), or

possibly the Ti record from the Cariaco Basin (Haug et al.?) This would seem to be a more direct way to address the question.

Response: This is a good question, and we have two answers to this. It would be very interesting to do a changepoint analysis on the Lechleitner et al. (2017) record of ITCZ variability to compare to our records. F. Lechleitner has been kind enough to share the z-scores from that paper with us, and we would like to add this record to our manuscript along with the ENSO, PDO, and AMO records. As for the Cariaco Basin titanium record (Haug et al. 2001; 2003), while its original interpretation was that it reflected changes in the ITCZ movement, and therefore could be used as an analog for climate on the Yucatan Peninsula, it has been shown that there are distinct differences in the climate signals in the Cariaco Basin record and other Yucatan Peninsula proxies (see Medina-Elizalde et al., 2010), and so we thought it might not be the best record to compare to.

Comments: And I think the analysis of the Mann et al., 2009 reconstructions was a good approach. It is interesting though, because individual proxy records of some of these climate modes show results that seem to differ from the Mann et al., 2009 reconstruction. For example, the Laguna Pallcacocha, Ecuador data (Moy et al., 2002) seems to show positive (warm) phase ENSO between about 800-1100 AD, which would be consistent with southward displacement of ITCZ and drought on the Yucatan. Interestingly, I think (but I could be wrong) that the Mann et al., 2009 reconstruction is based in part on this dataset, but the point is that there is reliable proxy data (from individual sites) that records different activity than the large-scale reconstructions.

Response: This is interesting, and in fact this seems to support our conclusion that ITCZ displacement played a role in the TCP droughts, but that local factors may also have contributed. It is our understanding that the Mann et al. (2009) reconstructions are regional averages of temperature anomalies calculated from global gridded temperature anomalies, which were reconstructed using a global climate proxy network. Therefore, the Laguna Pallcacocha record (Moy et al., 2002) was a part of the Mann et

al. (2009) reconstructions, but as the averages are weighted, it seems to have played more of a role at different time periods in the reconstructions (see Figure 2 & Figure S6, Mann et al., 2009). So, it is likely that the numerous proxy records that went into making the ENSO reconstruction do have some local differences from the large-scale reconstruction. Looking more closely at the Laguna Pallcacocha record (Moy et al., 2002), the red intensity of the sediment appears to be a proxy for the frequency of warm ENSO events, but not necessarily the magnitude of ENSO events. However, Moy et al. (2002) does note that peak ENSO variability occurs at about 750 C.E. (just before the TCP), and begins to decline after that- perhaps that is related to the change in mean ENSO state observed around 900 C.E. in our analysis? It is perhaps beyond the scope of this paper to look more into that particular proxy record (as we chose to focus only on Yucatan Peninsula proxy records), but it does seems to support our theory of local effects may have contributed to the TCP droughts, rather than just large scale circulation patterns.

Comments: I understand why the charts in Fig 3a and b are plotted by "Index value", which I think is basically the number of the sample starting from the earliest one, but why are the reconstructions in Fig 6 plotted by age? I assume the Mann et al., 2009 reconstructions are annual (I haven't checked recently) but it seems inconsistent.

Response: You are correct that the reason for this difference is that the reconstructions all have annual resolution, and so 1 index value = 1 year (as opposed to the proxy records, where 1 index value may represent numerous years). If this is confusing, we can easily change the axes back to index values.

Comments: Finally, there are improvements that could be made to the figures/tables captions to make the manuscript easier to understand. For example, the caption for Figure 5 says that two locations had records that meet all 4 criteria, and that these are highlighted by red boxes, but there are three red boxes at three locations.

Response: Thank you for catching this typo- the caption should read "The three locations of the Yucatan Peninsula which had records meeting all four inclusion criteria for drought (highlighted in red boxes)".

Comments: Technical corrections: Line 115: "A couple of" is too casual – please reword. Line 722: There is something unclear about this figure caption... does the mean (top) need to be mentioned when there is already a caption for it? Line 778: Should this be a table and not a figure? Line 794: Typo "at for each".

Response: Line 115: "A couple of hypotheses..." will be changed to "Two hypotheses...". Line 722- We think you meant Line 772: You are correct, we used to have this as one figure but when we posted online, it became two. The first sentence on Line 772 was a relic of that; the first line of the caption will be changed to say "Example results graphs from the changepoint analysis (variance). Line 778: While this looks like a Table, it has been uploaded as a figure due to the graphics added. We will leave this up to the editors of Climate of the Past as to whether they prefer this be described as a table or a figure. Line 794: The typo will be corrected to "for each".

References: Acuna-Soto, R., Stahle, D.W., Therrell, M.D., Chavez, S.G., & Cleaveland, M.K. 2005. Drought, epidemic disease, and the fall of classic period cultures in Mesoamerica (AD 750-950). Hemorrhagic fevers as a cause of massive population loss. Medical Hypotheses, 65: 405-409.

Haug, G.H., Hughen, K.A., Sigman, D.M., Peterson, L.C., & Rohl, U. 2001. Southward migration of the Intertropical Convergence Zone through the Holocene. Science, 293: 1304-1308.

Haug, G.H., Günther, D., Peterson, L.C., Sigman, D.M., Hughen, K.A., & Aeschlimann, B. 2003. Climate and the collapse of Maya civilization. Science, 299: 1731-1735.

Lechleithner, F.A., Breitenbach, F.F., Rehfeld, K., Ridley, H.E., Asmerom, Y., Prufer, K.M., Marwan, N., et al. 2017. Tropical rainfall over the last two millennia: evidence for a low-latitude hydrologic seesaw. Nature: Scientific Reports, 7(45809): 1-9.

Mann, M.E., Zhang, A., Rutherford, S., Bradley, R.S., Hughes, M. K., Shindell, D., Ammann, C., Faluvegi, G., and Ni, F. 2009. Global signatures and dynamical origins of the Little Ice Age and Medieval Climate Anomaly. Science, 326: 1256-1260.

Medina-Elizalde, M., Burns, S.J., Lea, D.W., Asmerom, Y., von Gunten, L., Polyak, V., Vuille, N., & Karmalkar, A. 2010. High resolution stalagmite climate record from the Yucatan Peninsula spanning the Maya terminal classic period.

Moy, C.M., Seltzer, G.O., Rodbell, D.T., & Anderson, D.M. 2002. Variability of El Niño/Southern Oscillation activity at millennial timescales during the Holocene epoch. Nature, 420: 162-165.

Turner, B.L., II. & Sabloff, J.A. 2012. Classic Period collapse of the Central Maya Lowlands: Insights about human-environment relationships for sustainability. PNAS, 109(35): 13908-13914.

———————————————————

---

## Author Comment (AC3) · 8 Oct 2019

Comment: The problem that this paper tackles is an interesting one: there has been much debate about whether the so-called 'Terminal Classic' drought represents a coherent interval of climate change across the Yucatan Peninsula, and what dynamics may be responsible for the drought. This paper definitely has potential, but the authors should review additional relevant literature and reframe or expand some of their analyses.

Response: We would like to thank Referee #2 for taking the time to read our manuscript and for their suggestions. We will respond to each comment individually for clarity.

[Figure]

Comment: First, why does the study only focus on the Yucatan Peninsula? If the authors are interested in looking for evidence of ITCZ changes, we would expect to see changes in the Caribbean, Gulf of Mexico, Central America, and northern South America. We may also expect to see antiphased changes in records from places farther south in the Amazon. We also have records of tropical storms from the wider circum-Caribbean region (see the work of Jeff Donnelly's group at WHOI). Can we more rigorously test the idea that the ITCZ may have shifted in this interval by using additional proxy records? The claim of using of '23' proxy records is a little bit misleading because many of the proxy records are from the same sites, and therefore are not really independent datapoints.

Response: Following the reviewer recommendation we will clearly state "23 proxy records from X sites". This approach recognizes that each proxy type may record changes in moisture differently while also emphasizing the number of unique proxy records examined. The reviewer makes a great suggestion to consider evidence for broader changes in the ITCZ to support our suggestion that drought responses in the Yucatan were driven by changes in the ITCZ. While we think that looking at the regional response of ITCZ movement in this area would be an interesting area of study, proxy records from the Yucatan Peninsula were specifically chosen to identify periods of potential drought which may have been related to the collapse of the Maya Civilization. Therefore, including more proxy records would be beyond the scope of this study. However, we plan to expand our discussion of shifts in the ITCZ to incorporate the regional records suggested by the reviewer.

Comment: Line 121: Given that pollen is not necessarily a linear indicator of forest cover, it is possible that there could have been intensified deforestation at the Terminal Classic – I recommend checking the land use reconstructions of Kaplan et al. 2011 "Anthropogenic Land Cover Change scenario for the preindustrial Holocene" to see what the reconstruction looks like in this particular region.

Response: The Kaplan et al. (2011) reconstructions are quite interesting, and in fact

support the pollen study by Leyden (2002) that suggests that there was wide defor-estation on the Yucatan Peninsula 800 years prior to the TCP. The Kaplan et al. (2011) reconstruction suggests that nearly 60% of the land in Mesoamerica was cleared by the year 1 C.E. We will include this study in the discussion of potential drought mecha-nisms in our manuscript.

Comment: One factor that isn't really considered in this study is the timing of social change or site abandonment in the archaeological record – we know for a fact that this was not uniform across the Maya region - see for instance Aimers et al., 2007 – this could be discussed more in the paper.

Response: We agree, and it was also pointed out by Referee #1 that we could incorpo-rate a section discussing archaeological evidence of the timing of the Terminal Classic Period. We would like to add this to the discussion section of our manuscript, and we think this would also be a good place to include a discussion of evidence for the spatial differences in site abandonment across the Yucatan Peninsula.

Comment: The citations in this paper are not really up to date - A few other papers that already address some of the themes in this paper, in some cases with more detailed analyses of the climate dynamics and age models for each site, should be discussed and cited. -Bhattacharya et al., 2017 in Quaternary Science Reviews includes a de-tailed analysis of the timing of drought in multiple records accounting for age uncer-tainty, and analyzes the drivers of drought in comprehensive climate models. -Evans et al 2018 in Science used new measurements of gypsum hydration waters and lake level modeling to estimate large changes in precipitation at the Terminal Classic. The estimates stand in contrast to Medina-Elizalde and Rohling, 2012, which estimated a modest change in rainfall. -There is also an interesting discussion in Metcalfe and Bar-ron, 2015, which reviews an extensive dataset of proxy records from across Mexico and parts of the Caribbean and Gulf of Mexico. These should be incorporated into the discussion, and can provide pointers on additional proxy records to incorporate into the text.

Response: We agree that these papers should be added to our discussion, and this was pointed about in the short comment from Nicholas Evans. These papers will be cited and discussed in the text along with other relevant work discussed in our manuscript.

Comment: Line 430: El Nino events do increase winter rainfall in this region, but they actually decrease summer rainfall. This is because warm ENSO events generate an atmospheric Kelvin wave that dampens surface precipitation in much of the tropics – see Lintner et al., 2005, Journal of Climate. There is a delayed response in the following spring that enhances rainfall as a delayed response to ENSO.

Response: We agree- this imbalance in the timing of precipitation is noted in lines 434-438 of our manuscript.

Comment: Line 442: I am skeptical of the inferences of Knudsen et al about the inverse relationship of AMO precipitation and Yucatan rainfall – it runs counter to much of what we know about the dynamics of the region. See the work on the Atlantic Warm Pool by Wang et al., 2005 in Journal of Climate, as well as the work by Giannini et al. that is cited in this paper. Overall, the paper addresses a topic worthy of study – it just needs revisions to the text and the inclusion of a greater number of proxy records to fully test the hypotheses it sets forth.

Response: This is an interesting point, and when we looked into this further, it seems there are varying opinions on the effect of AMO specifically on the Yucatan Peninsula, which lies between an area with a positive response to AMO (the Caribbean) and an area with negative response to AMO (Midwest/Central United States and Northern Mexico). We found reference to a negative precipitation response over the Yucatan Peninsula (e.g. Curtis, 2008; Knudsen et al., 2011) and more generally to a positive precipitation response over the Caribbean (e.g. Giannini et al. 2000; Wang et al., 2005; Wu and Kirtman, 2010). It is possible that the local response on the Yucatan Peninsula to AMO is more complex than in the Caribbean, and perhaps even a seasonal

response. The study by Curtis et al. (2008) suggests that while there is a decrease in mean rainfall during a positive AMO, there is also an increase in extreme rainfall events on the Yucatan Peninsula. We thank the reviewer for pointing this out, and plan to add a more in depth discussion of the possible climate effects of the AMO, and what this means for the TCP droughts.

References:

Aimers, J.J. 2007. What Maya collapse? Terminal Classic variation in the Maya lowlands. Journal of Archaeological Research, 15: 329-377.

Bhattacharya, T., Chiang, J.C.H., & Cheng, W. 2017. Ocean-atmosphere dynamics linked to 800-1050 CS drying in Mesoamerica. Quaternary Science Reviews, 169: 263-277.

Curtis, S. 2008. The Atlantic multidecadal oscillation and extreme daily precipitation over the US and Mexico during the hurricane season. Climate Dynamics, 30: 343-351.

Evans, N.P., Bauska, T.L., Gázquez-Sánchez, F., Brenner, M., Curtis, J.H., & Hodell, D.A. 2018. Quantification of drought during the collapse of the classic Maya civilization. Science, 361(6401): 498-501.

Giannini, A., Kushner, Y., & Cane, M.A. 2000. Interannual variability of Caribbean rainfall, and the Atlantic Ocean. Journal of Climate, 13: 297-311.

Kaplan, J.O., Krumhardt, K.M., Ellis, E.C., Ruddiman, W.F., Lemmen, C., & Goldewijk, K.K. 2011. Holocene carbon emissions as a result of anthropogenic land cover change. The Holocene, 21(5): 775-791.

Knudsen, M.F., Seidenkrantz, M-S., Jacobsen, B.H., & Juiipers, A. 2011. Tracking the Atlantic Multidecadal Oscillation through the last 8,000 years. Nature Communications, 2(178): 1-8.

Leyden, B.W. 2005. Pollen evidence for climatic variability and cultural disturbance in

the Maya lowlands. Ancient Mesoamerica, 13: 85-101.

Chiang, J.H.C., & Lintner, B.R. 2005. Mechanisms of remote tropical surface warming during El Niño. Journal of Climate, 18: 4130-4149.

Medina-Elizalde, M., & Rohling, E.J. 2012. Collpase of Classic Maya Civilization related to modest reduction in precipitation. Science, 335: 956-959.

Metcalfe, S.E., Barron, J.A., & Davies, S.J. 2015. The Holocene history of the North American Monsoon: 'known knowns' and 'known unknowns' in understanding its spatial and temporal complexity. Quaternary Science Reviews, 120: 1-27.

Wang, C., Enfield, D.B., Lee, S-K., Landsea, C.W. 2006. Influences of the Atlantic Warm Pool on western hemisphere summer rainfall and Atlantic hurricanes. Journal of Climate, 19: 3011-3028.

Wu, R., & Kirtman, B.P. 2020. Caribbean Sea rainfall variability during the rainy season and relationship to the equatorial Pacific and tropical Atlantic SST. COLA Technical Report 298: 42 pp.

---

## Author Comment (AC4) · 8 Oct 2019

We would like to add an additional response to one of the comments suggested by N. Evans, which has come to our attention after our initial response.

1. Comments: The use of published age models and the assumption they are accurate (which is highly unlikely) is critical to the subsequent comparison of the data to changepoint analysis conducted on PDO, ENSO and AMO signals. As a bare minimum, Bayesian age analysis should be used to quantify the errors in the age models. Sites with <5 radiocarbon (or other) dates in the last 2000-year interval should not be used (see Bhattacharya et al., 2017).

[Figure]

Initial Response: Upon revising this paper, we could expand on our assessment of the age models by employing Bayesian age analysis to quantify age uncertainty. We agree that sites with fewer than five age dates have much more uncertainty than those with more age dates. We attempted to account for this by ranking each proxy record in terms of confidence in that record. We don't believe it is necessary to completely exclude them from the analysis, as the point of this paper was to look at the available proxy records for the Yucatán Peninsula as a whole. We could, however, make it more clear in our conclusions that these records should not be considered to have definitive evidence of drought.

Additional Response: While we agree that it is possible the published age models for the proxy records used in this study are not completely accurate (and that this will be further expanded on in our discussion of uncertainty and identifying droughts), we are not convinced that Bayesian age analysis of the uncertainties would add much to this analysis. In Blaauw (2010), it is noted that for low-resolution dated age models, using Bayesian modeling techniques for the age-depth model may not provide much value compared to classical age-depth modeling. An increase in uncertainty with Bayesian methods (Bacon) using fewer radiocarbon dates is also noted in Trachsel and Telford (2016). As many of the proxy records used in this analysis have low resolution (see Supplementary Info, Table S1), and in particular a low number of radiocarbon dates (see Figure 1 of our manuscript), we believe that our qualitative assessment of uncertainty in the proxy records, which assigns more uncertainty to records with lower sample resolution and fewer radiocarbon dates, is still valuable for the assessment of uncertainty in these proxy records.

References:

Blaauw, M. 2010. Methods and code for 'classical' age-modelling of radiocarbon sequences. Quaternary Geochronology, 5: 512-518.

Trachsel, M. & Telford, R.J. 2016. All age-depth models are wrong, but are getting

better. The Holocene, 27(6): 860-869.
* * *